# TileLang: Bridge Programmability and Performance in Modern Neural Kernels

**Lei Wang[1], Yu Cheng[1], Yining Shi[1], Zhiwen Mo[2], Zhengju Tang[1], Wenhao Xie[1]**
**Tong Wu[1], Lingxiao Ma[3], Yuqing Xia[3], Jilong Xue[3], Fan Yang[4], Zhi Yang[1],***

[1]Peking University   [2]Imperial College London   [3]TileAI   [4]Microsoft Research

`yangzhi@pku.edu.cn, tile-ai@outlook.com`

## Abstract

Achieving high performance in modern AI increasingly requires kernels co-designed with underlying hardware, but writing efficient kernels remains challenging due to hardware-level complexity and limited fine-grained control in compilers like Triton. In this paper, we introduce TileLang, a programmable tile-level system that provides explicit primitives for memory placement, data movement, and parallel scheduling. Using a unified fused tile-level dataflow graph (FTG), TileLang streamlines kernel development by unifying tile recommendation, which guides developers with hardware-aware defaults, and tile inference, which automates completion through constraint propagation. TileLang enables concise expression of a wide range of AI algorithms in fewer than 70 lines of Python, reducing code size by up to 85.5% compared with manual implementations. Our evaluation shows that TileLang delivers $1.08\times$–$10.58\times$ speedups over Triton on NVIDIA H100 ($3.02\times$ on average) and $1.01\times$–$11.56\times$ on AMD GPUs ($2.65\times$ on average), effectively bridging programmability and performance. Our code is available at `https://github.com/tile-ai/tilelang`.

## 1 Introduction

The rapid progress of modern neural networks has driven a growing demand for highly optimized compute kernels, particularly for memory-bound operations such as attention. In recent years, modern attention algorithms such as Multi-Head Attention(MHA) (Vaswani et al., 2017), Multi-Head Latent Attention (MLA) (Liu et al., 2024), Gated Query Attention (GQA) (Ainslie et al., 2023), and Linear Attention (Gu & Dao, 2023; Dao & Gu, 2024; Sun et al., 2023; Yang et al., 2024), increasingly demand fine-grained control over memory hierarchy, scheduling, and data movement to fully utilize hardware capabilities. However, existing systems like Triton (Tillet et al., 2019) lack programmable abstractions to support this level of control. For instance, FlashMLA relies on carefully pipelined computations and shared memory reuse, but Triton gives programmers no direct control over tile reuse or pipeline scheduling, restricting performance optimization.

As a result, developers often face a steep trade-off between achieving peak performance and maintaining programmability: *they must either manually write complex CUDA kernels or sacrifice significant performance due to abstraction*

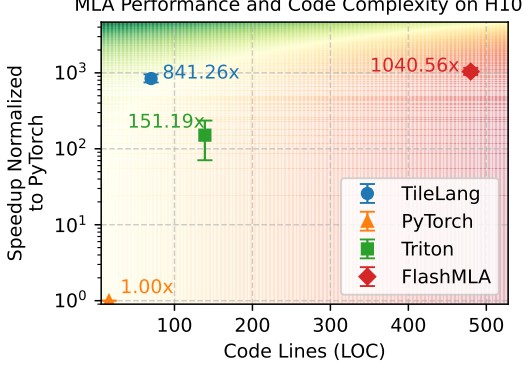

Figure 1: Performance vs. code size trade-off for MLA kernels on NVIDIA H100. Points closer to the top-left indicate better balance between performance and implementation simplicity. The annotated speedup values indicate performance gains over the PyTorch implementation.

---

*Corresponding author.

*mismatches*. As illustrated in Figure 1, the Triton implementation of MLA requires only 130 lines of code, whose convenience comes at a steep cost—its performance reaches only 14.2% of the hand-written CUDA version (DeepSeek, 2025) (∼500 lines) on NVIDIA H100 GPUs. Bridging the gap between programmability and performance requires addressing two key challenges. First, a programming model must give developers precise control over data movement and computation, enabling direct interaction with hardware resources. Second, a compiler must efficiently lower these high-level programs to GPU code, mapping abstractions onto hardware resources without adding programming complexity. Solving both challenges is essential to balance developer productivity with near-peak hardware performance on modern accelerators.

We introduce TILELANG, a *controllable programming system* for modern neural workloads. TILE-LANG provides programmable tile abstractions that let developers express and optimize low-level kernel behaviors in a high-level, composable way. Unlike existing compilers such as Triton, which rely on opaque optimization passes, TILELANG gives developers explicit control over memory, data movement, layout, and parallel execution. Specifically, developers can allocate buffers in different hardware memory levels (**alloc_shared**, **alloc_fragment**), orchestrate data transfers (**copy**), define custom memory layouts (**annotate_layout**, **use_swizzle**), and fine-tune parallelism and pipelining strategies (**Parallel**, **Pipelined**).

Under programmable tile abstractions, TILELANG programs can be represented as a unified fused tile-level dataflow graph (FTG). By operating on this FTG, TILELANG enables fine-grained reasoning and optimization of AI kernels, guiding developers from high-level design choices to fully specified, hardware-efficient kernel configurations. It introduces two complementary techniques. First, *tile recommendation* analyzes the FTG along with partially specified configurations to provide hardware-aware defaults for tile shapes, memory placement, and warp partitions, offering developers high-quality starting points that can be accepted, adjusted, or further tuned. Second, tile inference propagates shape and layout constraints across the FTG to complete the remaining configurations based on the partially annotated operators. It also automatically aligns buffer shapes, layouts, and memory allocations both downstream and upstream. This design blends flexible user control with automated optimization, yielding efficient kernels with far less manual effort.

As shown in Figure 1, TILELANG achieves on average 5.56× the performance of Triton and approaches the hand-written CUDA version in performance, while requiring less than 16% of the code size of the manual kernel and even fewer LOCs than Triton. This highlights TILELANG's ability to attain a more favorable balance between programmability and performance, offering both high efficiency and low development effort. We also implement other modern AI kernels—including Dequantize Matmul (Wang et al., 2024), Multi-Head Attention (MHA) (Vaswani et al., 2017), and Block-Sparse Attention (BSA) (Guo et al., 2024). Despite its deliberately streamlined interface, TILELANG achieves state-of-the-art throughput across heterogeneous GPUs, delivering speed-ups of up to 10.59× over Triton on an NVIDIA H100 and 11.56× on an AMD MI300X (AMD, 2024).

Our contributions are twofold: (1) programmable tile abstractions that let developers directly control and interact with hardware; and (2) tile recommendation and inference that guide developers with hardware-aware defaults and automatically complete configurations over a unified FTG graph. We believe TILELANG improves both the productivity and performance of modern AI kernel development.

## 2 RELATED WORK

**AI kernel programming and optimizations.** To simplify the development of AI kernels, libraries like FlashAttention-3 (Shah et al., 2024), CUTLASS (NVIDIA, 2019), and ThunderKittens (Spector et al., 2025) rely on manual or template-driven designs. Triton (Tillet et al., 2019) provides a high-level Python DSL but restricts control over critical performance paths. Gluon (OpenAI, 2025) is built on Triton DSL and exposes lower memory hierarchies like shared memory and registers. Helion (PyTorch, 2025) works as a higher-level DSL and is designed to compile down to Triton. Cypress (Yadav et al., 2025) introduces a task-based programming model with sequential semantics. Tilus (Ding et al., 2025) is another Python DSL for GPU programming, designed with thread-block-level granularity and tensors as the core data type. Mojo (Godoy et al., 2025) combines Python's interoperability and CUDA-like syntax to build performance-portable HPC science kernels. Frameworks such as PyTorch (Paszke et al., 2019), Graphene (Hagedorn et al., 2023), MLIR (Lattner et al., 2021), and Welder (Shi et al., 2023) take a compiler-centric approach. Unlike these works, TILELANG is a tile-level programmable language that automates layout and low-level configuration

while giving users fine-grained control. Its flexible tile programming abstraction can help researchers obtain kernels for a broad range of AI operations, and enable advanced optimizations like software pipelining (Cheng et al., 2025) and warp specialization (Huang et al., 2023).

**Cost modeling.** TANGRAM (Gao et al., 2019) optimizes dataflow across scheduling layers, along with a performance modeling tool extended by SET (Cai et al., 2023) with Resource Allocation Trees. KPerfIR (Guan et al., 2025) adds instrumentation for profiling and pipeline reordering in Triton. ML-based predictors like Path Forward (Li et al., 2023) and NEUSIGHT (Lee et al., 2025) also exist. In contrast, TILELANG's tile-level analytical cost model uniquely captures both computation and data movement at tile granularity, supporting fusion-aware scheduling with high accuracy and usability.

An extended discussion of related work is provided in Appendix A, covering classic tensor-level IRs (e.g., XLA (Google, 2019)), polyhedral compilation (Griebl et al., 1998; Zhao et al., 2021), loop-scheduling systems such as Halide (Ragan-Kelley et al., 2013), the TVM stack (Chen et al., 2018), CUTLASS (NVIDIA, 2019), and TaichiLang (Hu et al., 2019; 2020; 2021).

## 3 PROGRAMMING MODEL

### 3.1 TILE LANGUAGE

**Tile declarations.** TILELANG elevates a *tile*—a hyper-rectangular slice of a tensor—to a first-class citizen. A tile may be owned by a warp, a thread block, or any programmer-defined parallel unit, and can be reshaped or re-partitioned at compile time. In the `FlashMLA` kernel, the global matrices are consumed in tiles whose extents are parameterized by **block_H**, **block_N**, and related symbolic sizes. The **T.Kernel** structure establishes the kernel's launch configuration (e.g. **bx**, **by**, and the thread count), enabling both index derivation for each thread block and subsequent compiler analyses such as memory-access coalescing and loop tiling.

**Tile placement.** A distinguishing feature of TILELANG is the ability to map every tile buffer to a concrete level of the target accelerator's memory hierarchy via user-visible intrinsics, rather than relying on opaque compiler heuristics. **T.alloc_shared** reserves storage in low-latency, software-managed shared memory on NVIDIA GPUs (or an architecturally analogous space on other devices). **T.alloc_fragment** places accumulator tiles in the register file. Although registers are scarcer than shared memory, their single-cycle latency is indispensable for performance-critical reductions. During compilation, a layout-inference pass distributes these register tiles across threads while respecting register-pressure constraints and bank conflicts.

**Tile operators and schedulable primitives.** Table 1 in Appendix C showcases the representative subset of core building blocks that orchestrate computation and movement among tiles. Fundamental operators (**T.copy**, **T.gemm**, **T.reduce**) act on tile operands directly, allowing the programmer to express dense linear algebra, pointwise transforms, and reductions without resorting to scalarized loops. Orthogonal *scheduling primitives* expose fine-grained control over parallelism (**T.Parallel**), pipelining (**T.Pipelined**), and memory layout (**T.annotate_layout**, **T.use_swizzle**).

### 3.2 TILELANG PHILOSOPHY

**Tile-level tradeoff.** The system adopts tiles as the central abstraction because this granularity provides a practical balance between portability and performance. TILELANG models the GPU memory hierarchy and the major compute and data-movement units, exposing tile size, memory placement, warp partitioning, memory layout, and software pipelining as tunable dimensions. This design enables hardware-aware specialization on both NVIDIA and AMD while preserving a unified programming model. Remaining tradeoffs lie below the tile level, where extremely fine-grained hardware behavior cannot be captured through a stable and portable API.

**Novel tile abstraction.** Unlike prior systems where a "tile" is essentially a manually managed shared-memory buffer, TILELANG treats tiles as first-class IR constructs with explicit semantics for indexing, data movement, reuse, and pipelining. This makes tile behavior compiler-visible and supports systematic analysis and transformation. Consequently, TILELANG differs not only in surface syntax but also in the underlying IR, which enables principled optimization at tile granularity.

### 3.3 A Flash Multi-Head Latent Attention Example

By fusing high-level expressiveness with architecture-aware orchestration, TILELANG succinctly captures sophisticated AI algorithms such as FlashMLA (Liu et al., 2024) while fully harnessing the performance envelope of modern GPU architectures. Figure 2 illustrates TILELANG's developer–compiler co-optimization model: the *developer* specifies key decisions—such as tile configuration, launch grid (**block_H**, **block_N**), buffer placement (**T.alloc_shared**, **T.alloc_fragment**), swizzled layouts (**T.annotate_layout**), and warp-level collaboration (**T.Parallel**). The *compiler* then infers the remaining low-level details, including latency-hiding pipelines, conflict-free memory layouts, and instruction selection for peak hardware performance. To balance flexibility with automation, TILELANG offers two developer-facing facilities. First, *tile recommendation* (Sec. 4.2) supplies hardware-aware defaults that serve as high-quality starting points. Second, *tile inference* (Sec. 4.3) analytically propagates user-provided or recommended hints to complete the schedule and guarantee consistency. Working in concert, these facilities deliver near-optimal performance with limited manual tuning.

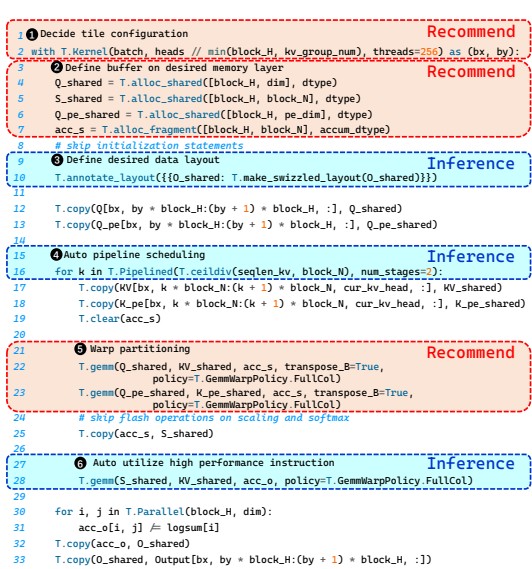

Figure 2: FlashMLA TILELANG kernel example

## 4 Scheduling Guidance and Automation

### 4.1 Two-stage framework

**Optimization space.** High-performance kernel design in TILELANG begins with a tile-level program, represented as a fused tile-level graph (FTG) capturing dataflow and tiling structure—each node represents a tile operator and each edge encodes a data dependency. By operating on this unified graph, TILELANG exposes and reasons about hardware-aware optimizations across six key dimensions: *tile size* (affecting shared memory and register usage), *memory placement* (selecting appropriate memory scope), *warp partitioning* (how threads collaborate and bind within a block), *memory layout* (how tile data is organized across memory levels), *software pipelining* (overlapping compute and data transfer, e.g., via TMA), and *tensorization* (mapping operations to CUDA or Tensor Cores).

**Tile recommendation and inference.** To efficiently explore the optimization space, TILELANG adopts a unified two-stage workflow over the FTG. In the first stage, *tile recommendation* analyzes the FTG to provide hardware-aware defaults for partially annotated operators, covering dimensions such as initial tile shapes, memory placement, and warp partitioning (Section 4.2). These recommendations shape the memory footprint, compute partitioning, and thread collaboration, providing high-quality starting points. In the second stage, leveraging the context from recommendation, *tile inference* propagates constraints through the FTG, automatically inferring the remaining configuration, including tile size, memory layout, software pipelining, and tensorization. It ensures consistency, compatibility, and hardware efficiency (Section 4.3). Together, these stages unify developer guidance and automated completion: recommendation narrows the design space with informed hints, while inference finalizes fully specified, hardware-efficient kernels with minimal manual effort.

In TileLang, the FTG defines the division of labor between developer control and system automation. Developers specify the FTG by composing tile operators; tile-level annotations such as tile sizes, memory placement, warp partitioning, and tensorization are optional. Given an FTG, TILELANG's optimization pipeline performs tile recommendation and inference, propagating shape, layout, and memory constraints to complete missing details. This design lets developers concentrate on describ-

ing computation while the system automatically finalizes and optimizes low-level configuration, supporting both fully automated and hint-guided usage.

**Running example.** Taking MLA as an example (Figure 2), TILELANG first performs tile recommendation as illustrated in Figure 3. Tile operators in the FTG expose tunable parameters—such as tile size, memory placement, and warp-partitioning strategies—serving as the user interface for these optimization knobs. For instance, in the first `T.gemm` operator (Figure 3 ❶), memory placement annotations specify Q and KV tiles in shared memory, while S resides in registers. The S tile is further partitioned across columns using the "`policy=FullCol`" warp-partitioning strategy. These decisions directly shape the memory footprint and influence data access patterns across the FTG. The cost model analyzes the FTG to estimate memory traffic, guiding the search toward configurations that minimize data movement.

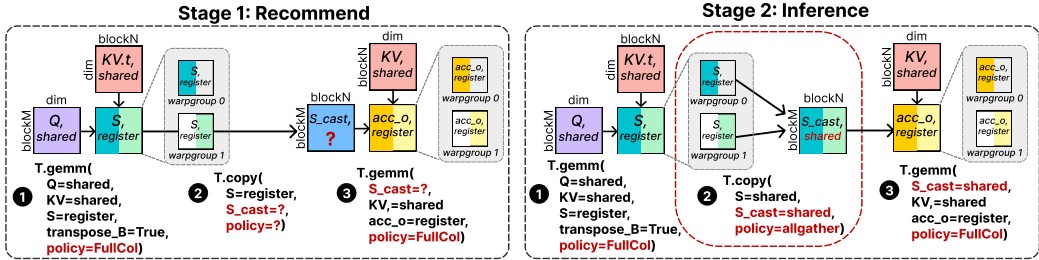

Figure 3: Two-stage workflow of optimizing MLA example.

Tile inference completes the configuration by operating over the FTG. For example, once S (output of the first `T.gemm` in Figure 3(❶) and S_cast (input of the second `T.gemm` in Figure 3❸) are fixed in location, shape, and partitioning in the first step, inference automatically determines the tile placement and partitioning (e.g., all-gather or scatter) of **copy** (Figure 3❷) in the second step, ensuring consistency without manual effort. Beyond copy decisions, inference also derives memory layouts by mapping multi-dimensional indices to physical addresses, explicitly considering vectorization, coalescing, and bank conflicts. Finally, it automates software pipelining and tensorization, ensuring that the resulting kernel configuration is efficient on the underlying hardware. TILELANG also provides platform-specific recommendations and inference (see Appendix D).

## 4.2 TILE RECOMMENDATION

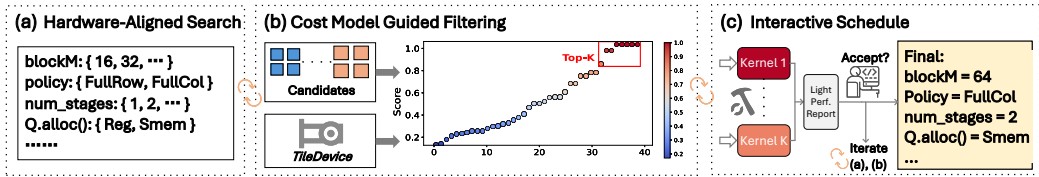

Figure 4: Tile recommendation with cost model. In (b), we show a scatter plot of candidate schedules. The x-axis orders candidates by their $\frac{1}{\text{predicted latency}}$, and the y-axis shows their normalized scores defined as $\frac{\text{latency of best candidate}}{\text{latency of current candidate}}$ which lies in the range (0,1].

**Roofline-based cost model.** As outlined in Figure 4, TILELANG uses a static roofline-based cost model to evaluate candidate configurations, including tile shapes, memory placement strategies, and warp partitioning. Each FTG under a given configuration is lowered into an IR that explicitly encodes compute and memory access patterns per tile. From this IR, the model statically estimates execution time as:

$$Time = \max_{i,j} \left( \frac{\text{MemoryTraffic}_i}{\text{Bandwidth}_i}, \frac{\text{Computation}_j}{\text{Performance}_j} \right) + t_{\text{intrinsic}}, \quad (1)$$

where $i$ indexes memory hierarchy levels (e.g., HBM, L2, L1), $j$ indexes compute unit types (e.g., tensor cores, CUDA cores, SFUs), and $t_{\text{intrinsic}}$ accounts for inherent overheads such as kernel launch latency and loop prologue/epilogue costs. The roofline formulation assumes perfect overlap between computation and memory transfers, providing a tight upper bound for rapid evaluation without runtime profiling.

Based on the cost model, TILELANG generates actionable recommendations for kernel tuning, including tile shapes, memory placement, and warp partitioning. These recommendations form an interactive baseline: developers can accept, adjust, or iteratively refine them across multiple rounds. This human-in-the-loop workflow balances automation with expert insight, slashing tuning effort while preserving full design control.

**Tile size.** TILELANG presents a ranked shortlist of tile shapes that are multiples of the device's native tensor-core fragments and respect register and shared-memory limits. Each candidate shows predicted arithmetic intensity, memory traffic, and roofline utilisation. Developers can accept the top choice, pin alternatives for later benchmarking, or adjust dimensions manually.

**Memory placement.** Given a chosen tile shape, TILELANG enumerates legal bindings of operands and temporaries to registers or shared memory, flagging options that exceed capacity. Each binding includes estimated pipeline stalls and effective bandwidth, letting developers quickly explore trade-offs and commit or refine placements.

**Warp partition.** To ensure sufficient thread-level parallelism, TILELANG proposes warp partitions that evenly cover the output tile and match the SM topology. With predicted occupancy and compute–memory overlap, developers can select, benchmark, or override, retaining full control while benefiting from data-driven guidance.

### 4.3 TILE INFERENCE

**Layout inference.** While memory placement and computation partitioning in Section 4.2 decide where tensors reside and how computation is split, layout inference determines how multi-dimensional indices are converted into physical memory addresses—taking into account vectorization, memory coalescing, and bank conflict avoidance. In other words, layout is not about which memory scope is used, but how data is accessed within that scope. Once placement and partitioning are fixed, the system can then infer an appropriate layout to ensure efficient low-level memory access.

TILELANG supports high-level indexing into multi-dimensional arrays (e.g., `A[i, k]`), which is eventually lowered to physical memory addresses through a hierarchy of abstractions. At the physical level, layouts are modeled as linear address expressions of the form $\sum_i y_i s_i$, where $y_i$ is the index along dimension $i$, and $s_i$ is its stride. To capture such mappings, TILELANG introduces a composable `Layout` algebra based on *IterVar*—a loop iterator that carries range and stride information. This allows layout transformations (e.g., transposes) to be expressed as algebraic mappings, such as `lambda i, j: (j, i)`. Formally, a layout becomes a function $f : \mathbb{K}^n \to \mathbb{K}^m$, converting high-level indices into memory addresses. Additionally, TILELANG defines `Fragment` layouts—a specialized extension where $f : \mathbb{K}^n \to \mathbb{K}^2$, mapping each index to a thread's register ID and its local offset. This enables precise modeling of intra-thread register allocation. Although a buffer of size $N$ theoretically allows $O(N!)$ memory layouts, the set of feasible layouts is significantly constrained by hardware. Global memory prefers coalesced access, shared memory requires bank conflict avoidance, and Tensor Core instructions impose strict layout requirements. To explore these constraints, TILELANG employs a greedy strategy that derives valid layouts by enforcing layout rules on selected tile operators.

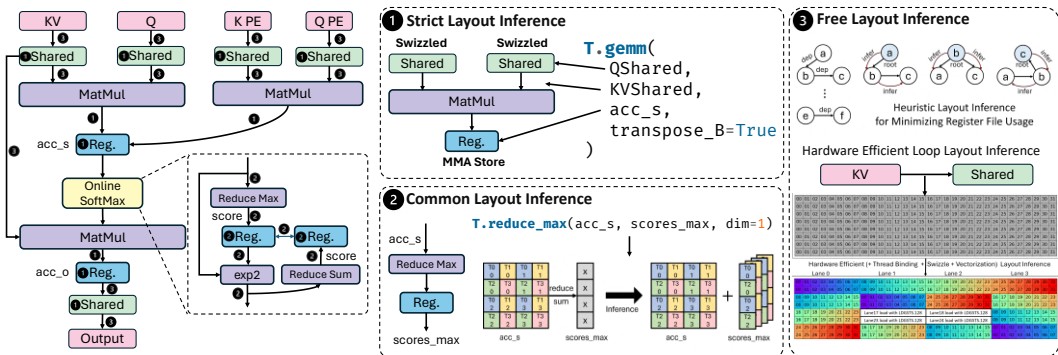

Figure 5: Layout Inference mechanism in TILELANG

We propose a hierarchical layout inference algorithm that operates over an FTG. As illustrated in Figure 5, FlashMLA can be represented as a FTG, where nodes are tile-level operators (e.g., `matmul`, `softmax`) and edges encode data dependencies. The graph captures how Q, K, and V tiles are loaded into shared memory, attention scores computed and normalized in registers, and final outputs written back. This structure makes memory movement and parallelism explicit, enabling layout inference and efficient scheduling.

Our goal is to synthesize memory layouts that optimize low-level execution efficiency while preserving high-level tensor semantics. The inference process is modeled as a constraint propagation algorithm (Algorithm 1 in Appendix E) that iteratively traverses the FTG and incrementally refines the layout mapping $\mathcal{L}$ until convergence. As illustrated in Figure 5, the algorithm integrates three complementary inference strategies: (1) Strict Layout Inference (Figure 5❶) enforces operator-specific constraints for hardware-sensitive primitives such as tensor core `GEMM`, including swizzled shared memory layouts and MMA-aligned register allocations; (2) Common Layout Inference (Figure 5❷) propagates layout decisions through structurally aligned operators (e.g., reductions), ensuring consistent thread bindings and register reuse; and (3) Free Layout Inference (Figure 5❸) handles the remaining unconstrained layouts by partitioning them into subgraphs via connected component analysis. For each subgraph, the partitioning scheme with the lowest register usage is selected. This step also determines the loop layout using the hardware cost model, which specifies thread binding and vectorization length to maximize memory coalescing and minimize bank conflicts. This unified inference pipeline supports composable, performance-portable layout generation and seamlessly bridges high-level loop indexing with low-level memory organization.

**Pipeline inference.** TILELANG automatically infers a pipelined schedule from a sequential program. As shown in Figure 6 (a), operations like **copy** and **gemm** are overlapped to increase parallelism. The system analyzes dependencies in the FTG and generates a structured pipeline that preserves execution correctness, exposing only a single `num_stages` parameter to users. Additionally, TILELANG applies Warp Specialization to fully exploit asynchronous copy instructions on Hopper GPUs, inserting synchronization barriers where necessary to maintain correct data dependencies. The detailed inference procedure is described in Appendix F.

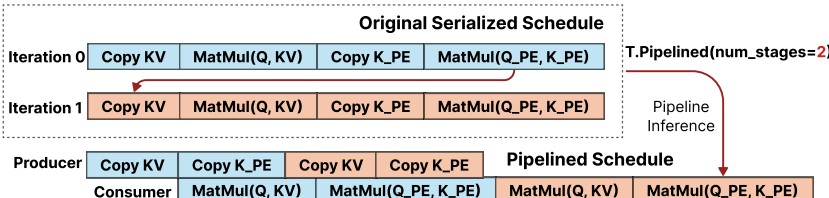

Figure 6: Pipeline Inference mechanism in TILELANG

**Instruction inference.** In TILELANG, while low-level hardware instructions such as `dp4a` or `mma` can be manually invoked via source injection or inline PTX (NVIDIA, 2021), choosing the most appropriate instruction based on input shapes and data types can be challenging. To address this, TILELANG integrates with high-level Tile Libraries like NVIDIA's **cute** (NVIDIA, 2019) and AMD's **ck** (AMD, 2025), which abstract hardware-specific details and automatically choose efficient instructions based on input configurations. These libraries expose standardized tile-based APIs (e.g., **tl::gemm_ss**), and TILELANG supports their invocation via a unified **T.call_extern** interface, simplifying development while ensuring performance portability.

## 5 EVALUATION

TILELANG is realized as a Pythonic DSL whose compiler lowers high-level tile programs to hardware-specialized kernels through a modular IR and code-generation pipeline. TileLang is implemented on top of the TVM backend, but our main contributions sit above TVM. TILELANG provides the tile abstraction, the FTG IR, and its own optimization passes, while TVM supplies the low-level code generation backend. As illustrated in Fig. 7, TILELANG adopts a five-stage compilation workflow: (1) tile-level code is written in a Python-based DSL; (2) the compiler translates the AST into the TileLang AST; (3) a FTG is constructed from the TensorIR; (4) a series of optimization passes in TileLang and TVM is applied; and (5) the optimized IR is finally lowered to CUDA, or other backends.

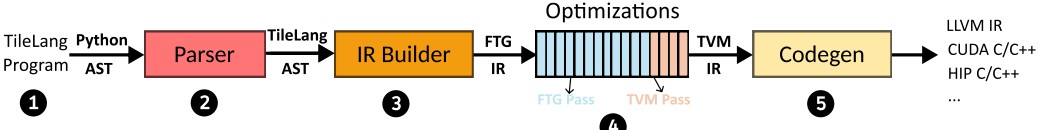

Figure 7: Overall TILELANG workflow.

## 5.1 EXPERIMENTAL SETUP

**Hardware platforms.** We assess the performance of TILELANG on two leading GPU architectures: NVIDIA and AMD, which dominate contemporary accelerator ecosystems. Our evaluation employs state-of-the-art hardware, including the NVIDIA H100 (80GB) (NVIDIA, 2023) and the AMD Instinct MI300X (192GB) (AMD, 2023). The NVIDIA H100 leverages CUDA 12.8, while the AMD MI300X utilizes ROCm 6.2.0. Both GPUs are benchmarked under the Ubuntu 20.04 operating system to ensure consistency in environmental configurations.

**AI kernels.** To evaluate system performance, we analyze nine representative operators: (1) GEMM, (2) fused dequantized GEMM ($W_{INT4}A_{FP16}$), (3) Attention, (4) Multi-Head Latent Attention, (5) Block Sparse Attention, (6) 2D Convolution, (7) Chunk Gated Delta Net, (8) Vertical Slash Sparse Attention, and (9) Attention Sink. Shape configurations are provided in Appendix N.

**Baselines.** Our comparative analysis considers the following baselines: (1) PyTorch Inductor—`torch.matmul` for GEMM, SDPA (PyTorch, 2023) for attention, and other operators compiled via Inductor; (2) Triton implementations, including GemLite (Mobius ML, 2024) and MLA from SGLang (Zheng et al., 2024); (3) ThunderKittens (TK) (Spector et al., 2025)—a template-based framework for high-performance AI kernels on NVIDIA GPUs; and (4) Highly optimized libraries, including CUTLASS (NVIDIA, 2019) and Composable Kernel (AMD, 2025) for GEMM, Marlin (Frantar et al., 2025) for dequantized GEMM, FlashAttention-V3 (Dao, 2023) for MHA, AITER (AMD, 2025) for MLA, and Block Sparse Attention (Guo et al., 2024) for sparse attention.

We evaluate kernel performance versus code complexity (Section 5.2) and present ablation results (Section 5.3), with cost model and tuning time analyses in Appendices G and H.

## 5.2 KERNEL PERFORMANCE

**Matrix Multiplication.** TILELANG achieves high performance with low code complexity across diverse GEMM configurations, demonstrating 1.18–1.40× speedup over PyTorch on NVIDIA H100, while maintaining competitive performance (0.94–1.05×) on AMD MI300X. It also delivers 1.08–1.43× speedup over Triton with minimal kernel code, enabled by automated inference that abstracts low-level hardware details such as TMA and pipeline scheduling. Compared with TK, TILELANG achieves 0.99–1.11× speedups while reducing code complexity by 77%. Its cost-model guidance and automated tile inference eliminate manual tuning. TK depends on curated CUDA templates, limiting it to NVIDIA GPUs, whereas TILELANG supports multiple hardware backends.

**Low-Bit Matmul.** For $W_{INT4}A_{FP16}$ GEMM, TILELANG achieves 1.35–3.81× speedups over PyTorch and up to 1.55× over Triton on H100, while outperforming the specialized Marlin kernel with far simpler code. On MI300X, it delivers on average 0.96 × over Triton. These gains arise because TILELANG exposes low-level memory, dequantization, and layout controls that are abstracted away in Triton.

**Convolution.** On H100, TILELANG achieves 1.24–1.79× and 1.10–1.97× speedups over PyTorch and Triton, respectively, with reduced code complexity. These gains come from its instruction inference mechanism, which maps data movement efficiently to TMA `im2col`. On MI300X, the improvements are even larger, reaching 1.29–6.80× over PyTorch and 1.02–3.10× over Triton.

**Flash Attention.** TILELANG achieves efficient attention computation with concise code across sequence lengths. On H100 and MI300X, it delivers 1.08–1.58× and 1.22–1.37× speedups over Triton, while matching the performance of FlashAttention-V3 (0.98× and 0.96× on average). These results stem from TILELANG's ability to infer and apply platform-specific partitioning and pipelining strategies that exploit specialized compute units. TILELANG achieves up to 1.10× speedup over TK while significantly reducing code complexity (from 185 lines to 66), highlighting its programma-

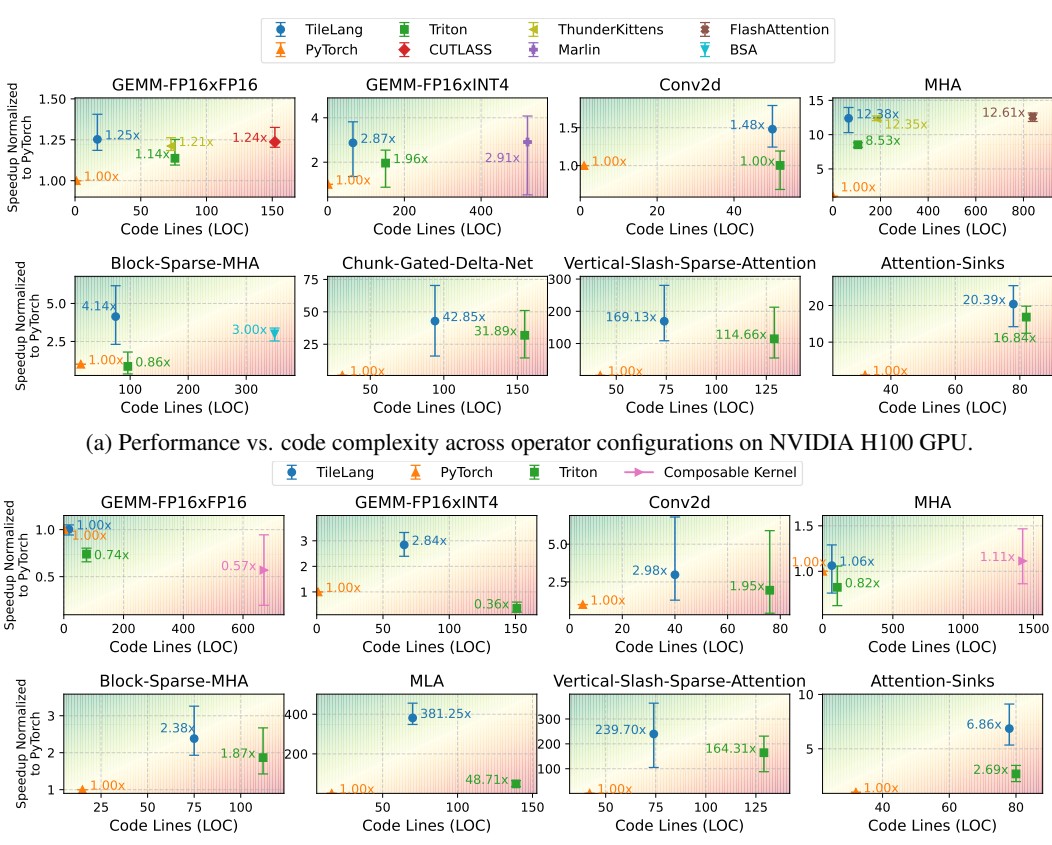

(a) Performance vs. code complexity across operator configurations on NVIDIA H100 GPU.

(b) Performance vs. code complexity across operator configurations on AMD MI300X GPU.

Figure 8: Performance and code complexity on an NVIDIA H100 GPU and AMD MI300X GPU. The y-axis denotes the speedup relative to PyTorch, while the x-axis indicates lines of code (LOC). Ideal solutions appear toward the top-left corner.

bility. By combining tile-level guidance with automated inference, TILELANG streamlines kernel development. This is particularly valuable for complex attention operators, where TK often requires extensive manual tuning of tiling, warp partitioning, layout, and pipelines.

**Flash MLA.** As shown in Figure 1, TILELANG achieves $4.06–10.59\times$ speedups over Triton on H100, with substantially reduced code complexity. It matches the latency of the specialized FlashMLA kernel while reducing code complexity by $6.86\times$. On MI300X, TILELANG delivers $5.64–12.97\times$ gains over Triton and slightly outperforms the hand-tuned ROCm library AITER ($1.05\times$). These improvements arise from warp specialization and automated TMA mapping.

**Block Sparse Attention.** TILELANG achieves acceleration of $3.42-7.87\times$ and $1.22-1.37\times$ over Triton with less code on H100 and MI300X, respectively. On H100, it matches BlockSparse (BSA) latency ($0.91–1.82\times$) while greatly reducing complexity. Implementing block-sparse MHA requires only adding two lines to the standard MHA code (Appendix O.5).

**Chunk Gated Delta Net.** On H100, TILELANG achieves $15.88–70.35\times$ speedups over PyTorch by fusing complex operations into a single kernel. Compared to Triton, it attains $1.10–1.45\times$ speedups with 39% fewer lines of code. These gains come from automated tile recommendation and inference, which optimize memory placement and partitioning for efficient hardware utilization.

**Vertical Slash Sparse Attention.** TILELANG delivers $108.55–280.41\times$ and $105.01–363.53\times$ speedups over PyTorch on H100 and MI300X, largely by fusing the sparse attention operation into a single efficient kernel. Compared to Triton, it achieves $1.16–1.97\times$ and $1.19–1.60\times$ speedups on H100 and MI300X, respectively, while cutting the code size by roughly half.

**Attention Sinks.** For attention with the sinks mechanism, TILELANG achieves 14.21–25.57× and 5.35–9.11× speedups over PyTorch on H100 and MI300X, respectively, enabled by TILELANG's FTG-based fusion into a single optimized kernel. Against Triton, it reaches 1.13–1.30× on H100 and 2.32–2.69× on MI300X. The attention-sink variant differs only slightly from standard MHA, showing that TILELANG readily supports diverse attention patterns with minimal effort.

## 5.3 ABLATION STUDIES

To help clarify what contributes to the speedups over the baseline, we perform an ablation study on FlashMLA as a representative example. Starting from a TILELANG version that uses manually crafted scheduling heuristics (TL-Heuristic), we progressively enable three components: (i) cost-model–guided tiling (+Tile), which improves the compute–memory ratio and cache use; (ii) cost-model–guided memory placement (+Alloc), which chooses efficient buffer locations and reduces register spilling; and (iii) warp partitioning (+Partition), which improves intra-warp load balance. Performance is measured at each stage relative to the Triton baseline.

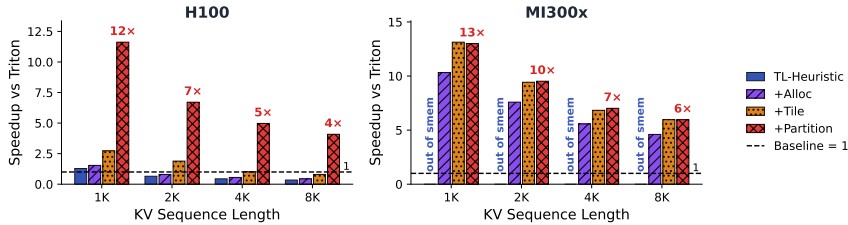

Figure 9: Ablation study for FlashMLA on both H100 and MI300X GPUs.

As shown in Figure 9, each of the evaluated optimizations provides measurable performance gains over the **Triton** baseline, validating their effectiveness. Our analysis also highlights architecture-specific behaviors (illustrated in Appendix D). On H100, tiling (+Tile) yields a modest speedup of 1.31 ×. Building on this, warp partitioning (+Partition) provides the dominant contribution, delivering an additional 4.34× improvement. On MI300X, allocation placement (+Alloc) serves as the primary optimization, achieving a 6.56 × speedup. When further combined with tiling (+Tile), the overall gain increases by another 1.75 × improvement.

## 5.4 COMPARE WITH MORE RECENT DSLS

We further compare TILELANG with recent DSLs including Helion, Gluon, and Tilus. Our evaluation covers the latest Hopper-supported implementations across GEMM, MHA, and Mamba-chunk-scan. Figure 10 reports the GEMM results, with additional results in Appendix L. On GEMM, TILELANG achieves 1.15 ∼ 1.62×, 1.52 ∼ 1.83×, and 1.87 ∼ 2.12× speedups over Helion, Gluon, and Tilus, respectively, while using fewer lines of code. These improvements largely stem from TILELANG's compiler-visible tile abstraction, which enables more structured optimization than existing DSLs. Note that Tilus is not yet fully optimized for Hopper features such as WGMMA and TMA, and on Ampere/Ada TILELANG is still slightly better (see Appendix L).

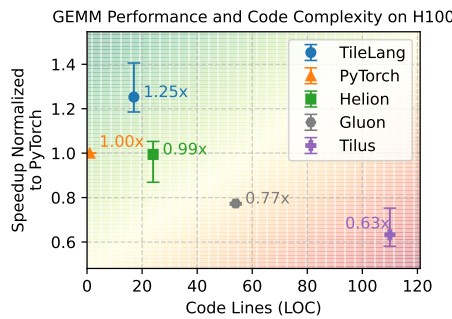

Figure 10: Performance and code-size comparison of GEMM kernels on H100 across recent systems.

## 6 CONCLUSION

TILELANG offers a controllable tile-level programming model with graph-based optimizations via tile recommendation and inference. By combining automated configuration with fine-grained developer control, it streamlines kernel development and delivers significant speedups. It enables rapid experimentation with emerging AI algorithms, such as custom attention, sparsity, and quantization. TILELANG also lowers barriers for systems-aware research across diverse hardware platforms.

## 7 REPRODUCIBILITY STATEMENT

We provide a detailed description of our experimental setup in Section 5.1. Operator shapes used in our benchmarks are drawn from widely adopted, real-world AI models (e.g., GPT-OSS, DeepSeek V3, Qwen3-Next). A list of these operator configurations is included in Appendix N, and the corresponding TILELANG code of kernels used in evaluation is provided in Appendix O. The system implementation and scripts for reproducing our experiments are publicly available, ensuring full reproducibility.

## 8 ACKNOWLEDGEMENT

We would like to thank all users and developers who have utilized and contributed to TileLang since its open-source release. We express special gratitude to DeepSeek for their pioneering adoption of TileLang in the development of new model algorithms. We also thank them for providing valuable feedback and significant code contributions to the community. This work was supported by the National Natural Science Foundation of China under Grant No. 62572009.

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

Our appendix is organized as follows:

# A    EXTENDED DISCUSSION OF RELATED WORK

**AI kernel generators**   For kernel generating and optimization, Ansor (Zheng et al., 2020) builds many kernel combinations by sampling programs from a hierarchical search space. PET (Wang et al., 2021) moves forward to partially equivalent transformations and automated correction for newly discovered kernels. TensorIR (Feng et al., 2023) generates new kernels by generalizing the loop nest representation used in existing machine learning compilers. Mirage (Wu et al., 2025) proposes a uniform representation of a tensor program at each level of the latest GPU compute hierarchy to find custom kernels. AKG (Zhao et al., 2021) leverages polyhedral schedulers to perform a much wider class of transformations to automatically generate kernels on NPUs. TileLang also utilizes multi-level programming interfaces and autotuning techniques based on tile-level cost model for kernel generation and optimization.

**Tensor-level IRs, The polyhedral model, and Loop synthesizers.**   Traditional approaches address program optimization at different abstraction levels: Tensor-level IRs (e.g., XLA (Google, 2019)) lower tensor programs via pattern-matched templates (e.g., LLVM, CUDA). Polyhedral models (Griebl et al., 1998) (e.g., TC (Vasilache et al., 2018)) automate affine loop transforms, mainly for DNN layers. Loop synthesizers (e.g., Halide (Ragan-Kelley et al., 2013)) generate loop nests guided by user-defined schedules. TILELANG targets a distinct programming model and control granularity. It differs fundamentally by introducing tiles as first-class programming units. It offers programmable control over fusion strategies, memory hierarchy, and parallelism. This enables developers to design fused kernels with both high performance and portability across hardware.

**TVM (Chen et al., 2018).**   TILELANG builds upon TVM's IR and arithmetic passes. However, unlike TVM's schedule-driven loop generation from high-level compute definitions, TILELANG offers explicit, tile-level programmability and control over memory, fusion, and parallelism. This enables much finer-grained kernel customization beyond what TVM can achieve. For instance, TVM cannot fully express advanced algorithms like FlashAttention (FA) or Multi-Head Latent Attention (MLA), which demand precise management of memory hierarchy and execution order—capabilities that TileLang supports.

**Warp Partition.**   Warp Partition (WP) is a key component of TILELANG's execution model, building directly on the tile abstraction. Given a specified tile size, WP allows further partitioning of the tile along each dimension across multiple warps. For example, consider a GEMM operation $C = A@B$,

where $A \in \mathbb{R}^{M \times K}$, $B \in \mathbb{R}^{K \times N}$, and $C \in \mathbb{R}^{M \times N}$. The output tile $C$ can be partitioned along either the $M$ or $N$ axis, corresponding to *full-row* or *full-column* warp-partitioning strategies, respectively. By giving users explicit control over warp partitioning, TILELANG enables fine-grained management of resources such as register usage within each warp. This, in turn, allows users to better control the performance of operations. Such flexibility is crucial for mapping computations efficiently to hardware, especially when optimizing diverse and performance-sensitive kernel workloads.

**CuTe library.** While both TILELANG and CuTe (NVIDIA, 2019) share this high-level goal, their underlying mechanisms differ: CuTe relies on shape/stride pairs, whereas TILELANG encodes the mappings using explicit arithmetic expressions. This arithmetic formulation offers advantages by more directly capturing index transformations and enabling more flexible, composable manipulations, allowing for clear definition and description in the DSL frontend.

**The roofline-guided cost model.** Several analytical modeling approaches have been proposed, such as the nested-loop-based modeling in Timeloop (Parashar et al., 2019), the data-centric representation in Maestro (Kwon et al., 2020). In contrast to these methods, our work leverages a tile-level programming abstraction, which naturally lends itself to a tile-centric cost model. This enables us to accurately capture both computation and data movement at the tile granularity, while maintaining simplicity and enhanced support for modeling operator fusion. This design strikes a balance between accuracy and usability, making it effective for guiding schedule selection without introducing excessive complexity.

**TaichiLang.** Although both Taichi (Hu et al., 2019; 2020; 2021) and TileLang target GPU workloads, they differ fundamentally in abstraction and intended use. Taichi provides a high-level, scalar-loop DSL with automatic parallelization, SNode-based data layouts, and strong autodiff support, making it well-suited for scientific computing and simulation. TileLang, by contrast, is designed for deep-learning kernels such as attention and GEMM, where peak performance requires explicit control of tile shapes, memory placement across shared/LDS and registers, multi-stage pipelining, warp specialization, and instruction selection (Tensor Core MMA or AMD MFMA). These capabilities are not directly expressible in Taichi's fully automatic model.

## B  MLA ALGORITHM

Instead of storing full-sized key and value matrices, MLA projects input token embeddings into a lower-dimensional latent space using a down-projection matrix:

$$\mathbf{z}_t = \mathbf{x}_t \mathbf{W}_{\text{down}}.$$

The latent vector $\mathbf{z}_t$ is then used to reconstruct the key and value representations:

$$\mathbf{k}_t = \mathbf{z}_t \mathbf{W}_{\text{up}}^K, \quad \mathbf{v}_t = \mathbf{z}_t \mathbf{W}_{\text{up}}^V.$$

To incorporate positional information, Rotary Positional Embedding (RoPE) is applied to the reconstructed keys and queries:

$$\mathbf{k}_t^{\text{rot}} = \text{RoPE}(\mathbf{k}_t).$$

Queries are also compressed using a similar process to reduce activation memory:

$$\mathbf{q}_t = \mathbf{z}_t^Q \mathbf{W}_{\text{up}}^Q.$$

MLA further enhances computational efficiency through a technique known as *matrix absorption*, which reorders matrix multiplications to optimize performance. This approach enables the key and value inputs to share the same latent representation $\mathbf{z}_t$, thereby reducing redundancy and memory usage. In the adopted configuration, MLA employs a single shared key-value (KV) head, with a head dimension of 512.

## C  PARTIAL LIST OF TILELANG PRIMITIVES

Table 1 illustrates the expressiveness of the TILELANG intermediate representation. To support efficient code generation across diverse hardware backends, TILELANG decouples the definition of algorithms from their optimization. The *Dataflow Centric Tile Operators* define the functional semantics of the workload (e.g., matrix multiplication, atomic updates), while the *Scheduling Primitives* expose critical optimization handles—such as loop pipelining ('Pipelined') and layout transformation ('annotate_layout')—to maximize hardware utilization and locality without altering the algorithmic correctness.

Table 1: A partial list of primitives supported by TILELANG.

| Dataflow Centric Tile Operators | | Scheduling Primitives | |
|---|---|---|---|
| `copy` | data movement among hierarchy memory. | `Parallel` | Parallelization of loop iterations over threads. |
| `gemm` | matrix multiplication on different GPUs. | `Pipelined` | Enables pipelining to overlap data transfers with computation. |
| `reduce` | reduction operator (e.g., sum, min, max) exploiting warp/block-level parallelism. | `annotate_layout` | Definition of custom memory layouts to minimize bank conflicts and optimize thread binding. |
| `atomic` | atomic operations to ensure thread-safe updates in shared or global memory. | `use_swizzle` | Improves L2 cache locality via swizzled access patterns. |
| Warp Specialization | | | |
| `barrier_arrive` | | Signals the arrival at a synchronization point (mbarrier) for producer/consumer coordination. | |
| `barrier_wait` | | Blocks execution until specific barrier conditions (e.g. transaction counts) are met. | |

## D  PLATFORM-SPECIFIC SCHEDULING

TILELANG also provides platform-specific recommendations and inference. Taking MLA as an example, we illustrate how TILELANG performs tile recommendation and inference based on the code shown in Figure 2.

On the H100, each SM features 228 KiB of shared memory and a 256 KiB register file, whereas the MI300X provides 64 KiB of Local Data Share (LDS) and a total of 512 KiB in registers. Given these architectural differences, TILELANG first recommends different tile configurations.

Table 2: Comparison of specifications between NVIDIA H100 SXM and AMD MI300X.

| Specification | NVIDIA H100 SXM | AMD MI300X |
|---|---|---|
| Clock Frequency | 1.83 GHz | 2.10 GHz |
| DDR Memory Bandwidth | 3.35 TB/s | 5.30 TB/s |
| L2 Bandwidth | 9.45 TB/s | 16.63 TB/s |
| L1/Shared Memory BW | 30.92 TB/s | 81.72 TB/s |
| Compute Units (SMs/CUs) | 132 SMs | 304 CUs |
| Shared Memory per SM/CU | 228 KiB | 64 KiB |
| Register File per SM/CU | 256 KiB | 512 KiB |
| Peak FP16 Performance | 989 TFLOPs | 1307 TFLOPs |

As shown in Figure 11, for memory placement, users may initially allocate the `Q` tile to shared memory on the MI300X. However, this approach fails due to the limited capacity of shared memory. TILELANG detects this constraint and instead recommends placing both `Q` and `acc_s` in registers. In contrast, on the H100, both tiles fit comfortably in shared memory and are placed there accordingly. For Software Pipelining, TILELANG disables pipelining on the MI300X to support larger tile sizes and reduce register pressure, whereas on the H100, pipelining is enabled to maximize pipeline overlap. Tile sizes are also adjusted accordingly to fit each platform's resource constraints. For Warp Partitioning, users may initially adopt a default policy for the two **gemm** operators, which often leads to sub-optimal performance. TILELANG addresses this by analyzing the underlying hardware and recommending platform-specific partitioning strategies, as illustrated in Figure 3. On the H100, both **gemm** operators use the `FullCol` scheme, partitioning `acc_s` and `acc_o` vertically to match the Tensor Core shape. In contrast, TILELANG applies a `FullRow` policy on the MI300X, partitioning tiles horizontally.

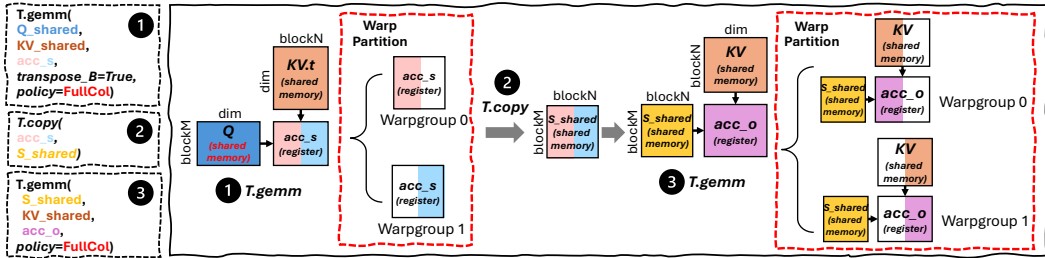

(a) The Warp Partition schedule recommended by TILELANG for MLA on H100.

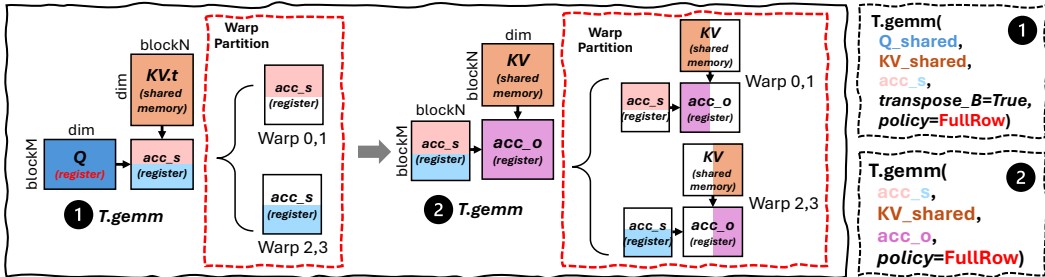

(b) The Warp Partition schedule recommended by TILELANG for MLA on MI300X.

Figure 11: Cooperative workflow between tile-recommendation and inference stages on NVIDIA H100 and AMD MI300X GPUs.

# E   LAYOUT INFERENCE ALGORITHM

**Goal and Scope.**   Algorithms 1 and 2 compute a hardware-aware, globally consistent layout mapping over a fused tile graph (FTG). Starting from partially annotated buffers and operator semantics, the pass infers concrete memory layouts (global/shared/register fragments), resolves aliasing, and returns both the final layout map $\mathcal{L}$ and the loop-level binding/predication maps (ForMap, PredMap) used by the loop-lowering routine in Algorithm 3. The inference explicitly respects coalescing, bank-conflict avoidance, and Tensor-Core-friendly register tiling, while minimizing register pressure for the remaining degrees of freedom.

**Three-Phase Inference.**   The core procedure (LayoutInference) executes in three stages:

- **Phase I: Strict constraints ("STRICT").** Each operator is visited once with RunInferStep at level "STRICT". This enforces hard constraints dictated by hardware-sensitive primitives (e.g., swizzled shared-memory layouts and MMA-aligned register tiles for GEMM). All layouts fixed here are recorded into $L_{\text{strict}}$ and treated as immutable thereafter.

- **Phase II: Common propagation ("COMMON").** A worklist over all operators drives fixed-point propagation. When an update materializes for a buffer, its users are enqueued. This phase spreads compatible, non-rigid constraints across the FTG until convergence, ensuring consistent thread bindings and compatible address formulas across producers and consumers.

- **Phase III: Free choices with register-cost minimization ("FREE").** Remaining unconstrained buffers are partitioned by connected components (w.r.t. uses and aliasing). For each component, the algorithm enumerates candidate "roots": it snapshots state, seeds inference from a root with level "FREE", and greedily extends to other members. Candidates that trigger a conflict (layout mismatch or iterator normalization errors) are discarded. Among feasible candidates, it selects the one minimizing total fragment registers via SumFragmentRegisters, then restores and commits the best snapshot. This realizes a lightweight, local backtracking that controls search while favoring low register pressure.

**Key Subroutines.**   RunInferStep constructs the per-operator context (target, thread bounds, current $L$, analyzer, and any out-of-bound info) and calls op.InferLayout(args, level)

**Algorithm 1** Hardware-Aware Layout Inference over Fused Tile Graph

```
 1: procedure LayoutInference(FTG, AnnotatedLayouts, Target)
 2:   (Ops, UseList, AliasGroups, ThreadVar, ThreadBounds, BufferOOB) = CollectFromIR(FTG, Target)
 3:   L = copy(AnnotatedLayouts)
 4:   L_strict = empty_map()
 5:                                                                   ▷ Phase I: Strict constraints
 6:   for i = 0 to |Ops| - 1 do
 7:       RunInferStep(i, "STRICT", false, L, L_strict)
 8:   for all (buf, lay) in L do
 9:       L_strict[buf] = lay
10:                                                                   ▷ Phase II: Common propagation
11:   q = queue_of_all_op_indices()
12:   FinishInferQueue("COMMON", L, L_strict, q)
13:                                                  ▷ Phase III: Free choices, minimize register cost
14:   for all comp in ConnectedComponents(Ops, UseList, AliasGroups) do
15:       best = (infinity, null)                                     ▷ (reg_cost, payload)
16:       for all root in comp.members do
17:           ops_bak = Snapshot(Ops)
18:           L_tmp = copy(L)
19:           ok = TryInferFromRoot(root, "FREE", L_tmp, L_strict)
20:           if ok then
21:               for all other in comp.members, other != root do
22:                   ok = ok and TryInferFromRoot(other, "FREE", L_tmp, L_strict)
23:           if ok then
24:               cost = SumFragmentRegisters(L_tmp)
25:               candidate = (cost, (Snapshot(Ops), L_tmp))
26:               best = MinByRegister(best, candidate)
27:           Restore(Ops, ops_bak)
28:       assert(best != null)
29:       (ops_snap, L_best) = best.payload
30:       ApplySnapshot(Ops, ops_snap)
31:       L = L_best
32:                                                                   ▷ Alias completion
33:   for all (var, buffers) in AliasGroups do
34:       if exists b in buffers such that L[b] is defined then
35:           ref = L[b]
36:           for all buf in buffers do
37:               if L[buf] is undefined then
38:                   L[buf] = ReshapeIfNeeded(ref, shape(buf))
39:   for all (buffer, dummy) in UseList do
40:       if scope(buffer) == "local.fragment" then
41:           assert(L[buffer] is defined)
42:   (ForMap, PredMap) = CollectLoopLayoutsAndPredicates(Ops)
43:   return (L, ForMap, PredMap)
```

---

**Algorithm 2** Core Subroutines for Layout Inference

---

1: **procedure** RunInferStep(op_id, level, update_queue, L, L_strict)
2: op = Ops[op_id]
3: args = {target, thread_bounds = ThreadBounds[op_id], layout_map = L, analyzer, buffer_oob = Buffer-OOB[op_id]}
4: updates = op.InferLayout(args, level)                            ▷ list of (buffer, layout)
5: **for all** (buf, lay_new) in updates **do**
6:     **if** buf in L **then**
7:         **if** scope(buf) == "local.fragment" and level != "STRICT" and buf not in L_strict **then**
8:             **if** FragmentContains(L[buf], lay_new) **then**
9:                 L[buf] = lay_new
10:                PropagateAlias(buf, lay_new, L, update_queue)
11:                **continue**
12:         assert(IsEqualLayout(L[buf], lay_new))
13:         PropagateAlias(buf, lay_new, L, update_queue)
14:     **else**
15:         L[buf] = lay_new
16:         PropagateAlias(buf, lay_new, L, update_queue)
17:         **if** update_queue **then**
18:             enqueue_all_users(buf)

19: **procedure** FinishInferQueue(level, L, L_strict, q)
20: **while** q is not empty **do**
21:     id = q.pop()
22:     RunInferStep(id, level, true, L, L_strict)

23: **procedure** PropagateAlias(src_buf, src_layout, L, update_queue)
24: **for all** sib in AliasGroups[src_buf.storage_var] where sib != src_buf **do**
25:     **if** shape(src_layout) == shape(sib) **then**
26:         tgt = src_layout
27:     **else**
28:         tgt = Reshape(src_layout, shape(sib))
29:     **if** sib in L **then**
30:         assert(IsEqualLayout(L[sib], tgt))
31:     **else**
32:         L[sib] = tgt
33:         **if** update_queue **then**
34:             enqueue_all_users(sib)

35: **procedure** TryInferFromRoot(root, level, L_tmp, L_strict)
36: success = false
37: **try**
38: RunInferStep(root, level, true, L_tmp, L_strict)
39: FinishInferQueue(level, L_tmp, L_strict, q)
40: success = true
41: **catch** LayoutConflict or NormalizeIterError
42: success = false
43: **return** success

44: **function** SumFragmentRegisters(L)
45: total = 0
46: **for all** (buf, layout) in L **do**
47:     **if** layout.kind == "Fragment" **then**
48:         total = total + product(layout.output_shape)
49: **return** total

---

---

**Algorithm 3** Loop Lowering: Binding, Vectorization, Predication

---

1: **procedure** ApplyLoopLayoutTransformations(ForLoop, ForMap, PredMap, thread_var)
2: loop_layout = ForMap[ForLoop]
3: parallel_loop =
        (not skip_thread_partition) and
        (not local_register_only(ForLoop)) and
        (not store_into_local(ForLoop))
4: **if** parallel_loop **then**
5:     ForLoop = PartitionLoop(ForLoop, thread_var, analyzer, loop_layout)
6: has_non_local = touches_non_local(ForLoop.body)
7: has_reducer = contains_reducer(ForLoop.body)
8: has_cast_ops = contains_nonreduction_cast_store(ForLoop.body)
9: **if** (has_non_local or has_cast_ops) and (not has_reducer) **then**
10:     ForLoop = VectorizeLoop(ForLoop)
11: **if** ForLoop in PredMap and parallel_loop **then**
12:     **return** IfThenElse(PredMap[ForLoop], ForLoop)
13: **return** ForLoop

---

to obtain layout updates. Updates are merged into $L$ with two safeguards: (i) for pre-existing buffers, strict equality is required unless the buffer is a register `Fragment` and the new layout *contains* the old one (per `FragmentContains`), in which case a safe refinement is allowed when not in "STRICT" and not locked by $L_{\text{strict}}$; (ii) every update triggers `PropagateAlias`, which reshapes to sibling shapes as needed and enforces alias-wise equality, enqueueing users when in worklist mode. `TryInferFromRoot` runs a guarded, queue-based inference seeded at a chosen root and catches `LayoutConflict`/`NormalizeIterError` to mark a candidate infeasible. `SumFragmentRegisters` accumulates the product of each fragment layout's output shape, serving as a proxy for total register footprint.

**Alias Completion and Validity Checks.** After the three phases, each alias group is revisited: if any sibling has a layout, the rest are filled by reshaping that layout to their shapes. The pass asserts that all `local.fragment` buffers used in the IR are defined. Finally, `CollectLoopLayoutsAndPredicates` summarizes loop-level binding decisions and out-of-bounds predicates into (`ForMap`, `PredMap`).

**Loop Lowering: Binding, Vectorization, Predication (Alg. 3).** Given a loop and (`ForMap`, `PredMap`), the lowering proceeds as follows:

- **Thread binding.** If the loop is parallelizable (not skipped, not purely local-register, and it touches non-local memory), `PartitionLoop` binds iterations to the hardware thread variable using the loop layout from `ForMap`, aligning with the previously inferred thread/block organization.

- **Vectorization.** If the loop body either touches non-local memory or performs non-reduction cast stores, and there is no reducer present, `VectorizeLoop` is applied. This realizes the vector length implied by the chosen layout, improving coalescing and matching hardware vector widths while mitigating bank conflicts.

- **Predication.** If the loop may encounter boundary conditions and is parallel, the loop body is guarded with `IfThenElse` using the predicate from `PredMap`, ensuring safe accesses without sacrificing parallel throughput.

**Discussion.** The division into STRICT/COMMON/FREE keeps the search tractable: rigid, hardware-mandated forms are locked first; compatible information is then propagated to convergence; and only the remaining degrees of freedom are explored via component-local, snapshot-and-choose search guided by a register-cost objective. Alias propagation guarantees storage-consistent address mappings, while fragment-aware refinement enables safe specialization of register tiling. The produced ($\mathcal{L}$, `ForMap`, `PredMap`) bridge high-level tile indices and low-level memory/thread organization, enabling performant, portable lowering across backends.

---

**Algorithm 4** Pipeline Inference

---
1: **procedure** PipelineInference($f$, NumStages, InitOrderMap, InitStageMap)
2:   OrderMap $\leftarrow$ copy(InitOrderMap)
3:   StageMap $\leftarrow$ copy(InitStageMap)
4:   **for all** serial loop $L$ in $f$.body **do**
5:     **if** $L \in$ OrderMap **and** $L \in$ StageMap **then**
6:       **continue**
7:     **if** $L \notin$ NumStages **then**
8:       **continue**
9:     $n =$ NumStages$[L]$
10:    root $= L$.body
11:    seq $=$ FlattenToSeq(root)
12:    Infos $\leftarrow [\,]$
13:    **for** $i = 0$ to $|$seq$| - 1$ **do**
14:      $(R, W, C) =$ RWCollect(seq$[i]$)
15:      Infos.push_back(StageInfo(R, W, $i$, C))
16:    $S =$ CollectCopyReads(Infos)
17:    PropagateProducers(Infos, $S$)
18:    ComputeLastUse(Infos)
19:    order_idx $= 0$
20:    **for all** $p$ in Infos **do**
21:      **if** FirstStage($p$) **and** $p$.last_use $\neq -1$ **then**
22:        **continue**
23:      $p$.order $=$ order_idx; order_idx $\leftarrow$ order_idx $+ 1$; $p$.stage $= n$
24:      **for all** $q$ in Infos **do**
25:        **if** FirstStage($q$) **and** $q$.last_use $= p$.original_idx **then**
26:          $q$.order $=$ order_idx; order_idx $\leftarrow$ order_idx $+ 1$; $q$.stage $= 0$
27:    **assert**(order_idx $= |$Infos$|$)
28:    $k =$ TailCopyCount(Infos)
29:    **if** $k > 0$ **and** $n \geq 2$ **then**
30:      **for all** $p$ in Infos **do**
31:        $p$.order $= (p$.order $+ k)$ mod $|$Infos$|$
32:        **if** not $p$.copy **and** not $p$.producer **then**
33:          $p$.stage $= p$.stage $- 1$
34:    orders $= [p$.order for $p \in$ Infos$]$
35:    stages $= [p$.stage for $p \in$ Infos$]$
36:    ApplySoftwarePipeline($L$, orders, stages, OrderMap, StageMap)
37: **return** (OrderMap, StageMap)

---

## F   Pipeline Inference Algorithm

**Goal and Scope.** Algorithm 4 computes a software-pipelined schedule for serial loops that expose staged data movement and computation. Given an input function $f$, an a priori upper bound on the number of pipeline stages `NumStages`, and optional initial order/stage annotations (`InitOrderMap`, `InitStageMap`), the pass produces a pair of maps (`OrderMap`, `StageMap`). For each eligible serial loop $L$ in $f$.body, the algorithm assigns (i) a total order index to every statement in the flattened loop body and (ii) a stage id in $\{0, \dots, n-1\}$, where $n =$ `NumStages`$[L]$, thereby enabling backend-specific software pipelining and overlapped execution of copies and compute.

**Loop Selection and Linearization.** The outer procedure `PipelineInference` first filters serial loops: if a loop $L$ already has entries in both `OrderMap` and `StageMap`, or if it lacks a stage budget in `NumStages`, it is skipped. For each remaining loop, the body is linearized into a sequence `seq` via `FlattenToSeq`, which yields a stable, single-pass order of statements. Each sequence element is then summarized into a `StageInfo` record containing its read set, write set, original index, and a Boolean flag indicating whether the statement performs a global-to-shared copy.

**Read/Write Classification and Copy Detection.** The helper `RWCollect` (Algorithm 5) traverses a statement and classifies its memory behavior into three components: a set of read regions $R$, a

---

**Algorithm 5** Core Subroutines for Pipeline Inference

---

1: **function** RWCollect(stmt)
2: $R = [\,]; W = [\,]; C = $ false; within = false; isg = false
3: **Visit**(stmt):
4:    **on** BufferStore($b$, idxs, $v$):
5:       $W$ += Region($b$, idxs); isg = false; **Visit**($v$);
6:       **if** isg **and** scope($b$) $\in$ {"shared", "shared.dyn"} **then** $C = $ true;
7:    **on** BufferLoad($b$, idxs):
8:       $R$ += Region($b$, idxs);
9:       **if** scope($b$) == "global" **and** not within **then** isg = true;
10:    **on** IfThenElse($c$, $a$, $b$):
11:       within = true; **Visit**($c$); within = false; **Visit**($a$);
12:       **if** $b$.defined() **then** **Visit**($b$);
13: **return** $(R, W, C)$
14:
15: **function** CollectCopyReads(Infos)
16: $S = $ set()
17: **for all** $p$ in Infos **where** $p$.copy **do**
18:    **for all** $r$ in $p$.reads **do**
19:       $S$.add($r$.buffer)
20: **return** $S$
21:
22: **procedure** PropagateProducers(Infos, $S$)
23: **for all** $p$ in Infos **where** $p$.copy **do**
24:    upd = true
25:    **while** upd **do**
26:       upd = false
27:       **for all** $q$ in Infos **where** not $q$.copy **and** $q$.original_idx $<$ $p$.original_idx **do**
28:          **if** exists $w$ in $q$.writes **with** $w$.buffer $\in S$ **then**
29:             $q$.producer = true; upd = true
30:             **for all** $r$ in $q$.reads **do**
31:                $S$.add($r$.buffer)
32:
33: **procedure** ComputeLastUse(Infos)
34: **for all** $p$ in Infos **where** FirstStage($p$) **do**
35:    **for** $i = p$.original_idx $+ 1$ to $|$Infos$| - 1$ **do**
36:       **if** exists $r$ in Infos[$i$].reads, $w$ in $p$.writes **with** $r$.buffer $= w$.buffer **and** MayConflict($r$.region, $w$.region) **then**
37:          $p$.last_use $= \max(p$.last_use, $i)$
38:
39: **function** TailCopyCount(Infos)
40: $c = 0; mn = |$Infos$|; mx = 0$
41: **for all** $p$ in Infos **do**
42:    **if** FirstStage($p$) **then**
43:       $c \leftarrow c + 1; mn = \min(mn, p$.order$)$
44:    **else**
45:       $mx = \max(mx, p$.order$)$
46: **if** $mn > mx$ **then**
47:    **return** $c$
48: **else**
49:    **return** $-1$
50:
51: **procedure** ApplySoftwarePipeline($L$, orders, stages, OrderMap, StageMap)
52: OrderMap[$L$] $\leftarrow$ orders
53: StageMap[$L$] $\leftarrow$ stages
54:
55: **function** FirstStage($p$)
56: **return** $p$.copy **or** $p$.producer
57:
58: **function** MayConflict($a$, $b$)
59: **return** Intersect(IntSet($a$), IntSet($b$)) $\neq$ Nothing

---

set of write regions $W$, and a Boolean flag $C$ for copy-like behavior. The visitor marks loads and stores according to buffer scope (e.g., `global`, `shared`, `shared.dyn`), and uses a simple state machine over the control-flow context (`within`, `isg`) to detect global loads that feed subsequent shared-memory stores. Whenever a `BufferStore` into a shared buffer is preceded by such a global read, the statement is classified as a *copy* ($C = \texttt{true}$), allowing the later stages to identify candidate prologue/epilogue moves for software pipelining.

**Producer Propagation and Lifetime Analysis.** Given the per-statement summaries, `CollectCopyReads` aggregates the set $S$ of buffers read by copy statements, which serves as the seed for producer discovery. `PropagateProducers` then iteratively walks backwards over the sequence to mark statements that *produce* any of the buffers in $S$ before a copy: whenever a non-copy statement writes to a buffer in $S$, it is labeled as a producer, and its own read buffers are added to $S$. This fixed-point propagation captures multi-hop producer chains that eventually feed global-to-shared copies.

Next, `ComputeLastUse` computes a conservative *last-use index* for each first-stage statement (copy or producer). For a given $p$, the pass scans later infos and checks whether any read region of a later statement may conflict with any write region of $p$, using `MayConflict` and an interval-set intersection test. The largest index where such a conflict occurs is recorded as $p$.last_use, providing a lifetime window that guides stage assignment and the positioning of prefetch-like operations.

**Stage Assignment and Tail Rotation.** With producer/copy labels and last-use information in place, `PipelineInference` assigns an initial order and stage for each info in a single forward pass. The core policy is that a first-stage statement $p$ that is *dead* within the steady-state (i.e., `FirstStage`$(p)$ and $p$.last_use $= -1$) participates directly in the final pipelined schedule; otherwise, such statements are skipped in this phase until their consumers are placed. For each selected $p$, the algorithm emits $p$ at the current order index with stage $n$ (typically the last stage), then searches for matching first-stage statements $q$ whose last use equals $p$.original_idx and places those $q$ immediately after $p$ at stage $0$. This yields an interleaving of first-stage and steady-state work that respects data dependencies and the inferred lifetimes. The invariant `order_idx` $= |\text{Infos}|$ is asserted at the end to guarantee that all statements receive a unique order.

To further improve the pipeline structure, `TailCopyCount` detects whether first-stage statements form a contiguous tail segment in the assigned order. It counts the number of first-stage infos $c$, and tracks the minimum order index among them ($mn$) and the maximum order index of non-first-stage statements ($mx$). If $mn > mx$, first-stage statements appear strictly after all other work, and the function returns $c$; otherwise, it returns $-1$. When a positive tail count $k$ is found and at least two stages are available ($n \geq 2$), the algorithm rotates the schedule by $k$ positions in a modular fashion and decrements the stage of non-copy, non-producer statements. Intuitively, this rotation shifts tail copies into the prologue while pulling steady-state computation earlier, yielding a more balanced pipeline across the $n$ stages.

**Map Materialization and Backend Interface.** After rotation, the final per-statement orders and stages are collected into arrays `orders` and `stages`, which are committed to the global maps via `ApplySoftwarePipeline`. For each loop $L$, `OrderMap`$[L]$ records a permutation of the flattened body, and `StageMap`$[L]$ records a stage id for each element in that permutation. These maps serve as the contract between the high-level pipeline inference and downstream backends: code generators can exploit the stage structure to schedule prefetches, overlaps of global-to-shared copies with compute, and explicit prologue/epilogue code, without re-running dependence analysis on the original IR.

**Discussion.** The pipeline inference algorithm deliberately decouples (i) classification of copy and producer statements, (ii) lifetime and conflict analysis, and (iii) stage-aware ordering and optional schedule rotation. The use of region-based read/write summaries and conservative `MayConflict` checks ensures correctness under aliasing and partially overlapping accesses. At the same time, the simple rotation heuristic (`TailCopyCount`) captures a common pattern in GPU kernels where global-to-shared transfers form a logical prologue or epilogue. By emitting (`OrderMap`, `StageMap`) instead of directly rewriting the IR, the pass remains backend-agnostic

while still exposing enough structure for aggressive software pipelining and latency hiding across diverse hardware targets.

# G  EFFECTIVENESS OF THE COST MODEL

TILELANG employs an analytical cost model to prune suboptimal candidates and prioritize high-potential ones. This approach yields schedules that match or closely approach the performance of the best results of brute-force search or exhaustive autotuning, while requiring orders of magnitude less tuning effort. For example, on GEMM-FP16×FP16 workloads derived from models such as LLaMA-70B, TILELANG prunes 95% of candidate schedules, retaining only the top 5%. Despite this aggressive pruning, it achieves on average 98.47% of the performance (in TFLOPS) of the best configurations found by exhaustive search, substantially reducing compilation time with only negligible performance loss. This also serves as an evaluation of the effectiveness of our roofline-based analytical cost model.

| $M$ | $N$ | $K$ | Predicted-TopX / Best (TFLOPS) |
|---:|---:|---:|---:|
| 512 | 1024 | 8192 | 100.0% |
| 512 | 12288 | 12288 | 99.9% |
| 512 | 28672 | 8192 | 98.7% |
| 2048 | 12288 | 49152 | 100.0% |
| 4096 | 1024 | 7168 | 100.0% |
| 4096 | 14336 | 14336 | 100.0% |
| 4096 | 28672 | 8192 | 99.6% |
| 8192 | 8192 | 28672 | 100.0% |
| 8192 | 28672 | 8192 | 100.0% |
| 16384 | 1024 | 7168 | 98.4% |

Table 3: Accuracy of our analytical cost model: predicted top-5% schedules retain over 98% of the best performance while pruning 95% of candidate schedules.

# H  TUNING TIME

As demonstrated in Table 4, TILELANG leverages a hardware-aware recommendation mechanism to efficiently automate the design of high-performance computational kernels. The system achieves average tuning durations of approximately 10 seconds across both NVIDIA H100 and AMD MI300X accelerators. For the most complex operations, extended tuning times average 13.69 seconds on H100 and 15.08 seconds on MI300X, reflecting the scalability of our approach under computationally intensive workloads.

We also conducted a direct tuning-time comparison with Ansor/AutoTVM and Triton for the GEMM and 2D convolution kernel on H100. TileLang and Triton are tuned with 20 configs, and the number of trials is set to 100 in Ansor. The results in Table 56 show that TileLang tunes markedly faster than both frameworks—especially vs. TVM Ansor, which requires much longer empirical search. This improvement comes from TileLang's first-class tile IR, which defines a far more structured optimization space, and from our cost-model–guided inference, which avoids large brute-force searches.

Table 4: Average Tuning Times for Different Operators

| Operation | GEMM | DequantGEMM | FlashMHA | FlashMLA | FlashBSA |
|---|---|---|---|---|---|
| H100 Time (s) | 9.05 | 9.15 | 13.48 | 13.69 | 13.46 |
| MI300 Time (s) | 10.99 | 11.10 | 14.67 | 15.03 | 15.08 |

Table 5: Comparison of Average Tuning Times for GEMM

| Operation | GEMM1 | GEMM2 | GEMM3 | GEMM4 |
|---|---|---|---|---|
| TileLang Time (s) | 11.81 | 11.78 | 14.59 | 14.31 |
| Triton Time (s) | 18.43 | 18.24 | 20.15 | 20.07 |
| Ansor Time (s) | 518.52 | 455.51 | 3007.00 | 4142.05 |

Table 6: Comparison of Average Tuning Times for Conv2D

| Operation | Conv2D1 | Conv2D2 | Conv2D3 | Conv2D4 | Conv2D5 | Conv2D6 | Conv2D7 | Conv2D8 |
|---|---|---|---|---|---|---|---|---|
| TileLang Time (s) | 11.56 | 17.49 | 17.65 | 17.41 | 19.18 | 18.87 | 17.50 | 17.21 |
| Triton Time (s) | 17.56 | 18.76 | 37.59 | 38.57 | 37.61 | 18.82 | 19.13 | 18.97 |

## I  MATMUL IMPLEMENTATION DIFFS: TVM VS. TILELANG VS. TRITON

TILELANG achieves significant code-size reduction through its fundamentally different tile abstraction. Instead of manipulating raw pointers, TILELANG represents tiles as first-class IR constructs, endowed with explicit semantics for indexing, data movement, and pipelining. High-level primitives such as `copy`, `gemm`, and `pipelined` enable the compiler to automatically perform address computation and pipeline orchestration. In contrast, Triton requires programmers to manually manage memory access via pointer and offset arithmetic, resulting in a lower level of abstraction.

Moreover, TILELANG supports maintaining execution context via `T.Kernel`, which obviates the need for developers to explicitly compute grid dimensions and launch kernels. This design choice further reduces code size by eliminating boilerplate associated with kernel invocation.

In TVM, matrix multiplication kernels are typically expressed through simple tensor-compute definitions. Performance optimization is achieved by applying hand-written schedules, which can transform the computation into a high-performance form. While this manual scheduling process may require tens to hundreds of lines of Python code, TVM also provides automatic scheduling mechanisms such as Ansor to explore schedule configurations. After scheduling, the tensor expressions are compiled and built into executable code.

However, TVM's scheduling-independent compute-expression abstraction has limited expressiveness for certain operators, such as FlashAttention and irregular sparse kernels. Automatic scheduling in TVM also faces challenges when dealing with a very large search space and when targeting new hardware backends with insufficient operator expressiveness. TILELANG addresses these limitations by employing a human-in-the-loop methodology to enhance expressiveness for complex and irregular workloads, and by leveraging a cost-model–driven scheduling approach to mitigate the search-space explosion issue.

## J  COMPARISON OF FTG IR AND TVM IR

Figure 13 (a) shows an FTG IR produced after the Pipeline Inference Pass, which explicitly materializes a software pipeline with a prologue (prefetch the first A/B tiles and clear the accumulator), a steady-state loop that overlaps compute and prefetch, and an epilogue (final compute and write-back). This IR emphasizes schedule semantics and readability: block tiling, shared-memory staging, and double buffering are expressed with concise, high-level primitives such as `T.Kernel`, `alloc_shared`/`alloc_fragment`, `T.clear`, `T.copy`, and `T.gemm`. The same semantics are then lowered to TVM IR in Figure 13 (b), which preserves the pipeline structure while making GPU execution details explicit: `blockIdx`/`threadIdx` bindings, scoped buffers in shared/local memory, unrolled initialization, asynchronous copies (e.g., `cp.async`) with commit/wait groups, and a target-specific GEMM call. In short, (a) captures a tile-wise schedule for readability and portability, whereas (b) exposes fine-grained GPU mechanisms for maximum control and performance, without changing the prologue–steady-state–epilogue structure.

**❶ TileLang**

```
1   @T.prim_func
2   def matmul_kernel(
3       A: T.Tensor((M, K), dtype),
4       B: T.Tensor((K, N), dtype),
5       C: T.Tensor((M, N), dtype),
6   ):
7       with T.Kernel(T.ceildiv(N, block_N), T.ceildiv(M, block_M)) as (bx, by):
8           A_shared = T.alloc_shared((block_M, block_K), dtype)
9           B_shared = T.alloc_shared((block_K, block_N), dtype)
10          C_local = T.alloc_fragment((block_M, block_N), accum_dtype)
11          T.clear(C_local)
12          for ko in T.Pipelined(T.ceildiv(K, block_K)):
13              T.copy(A[by * block_M, ko * block_K], A_shared)
14              T.copy(B[ko * block_K, bx * block_N], B_shared)
15              T.gemm(A_shared, B_shared, C_local)
16          T.copy(C_local, C[by * block_M, bx * block_N])
17
18  A, B, C: torch.Tensor
19  matmul_kernel(A, B, C)
```

**❷ TVM**

```
1   A = te.placeholder((M, K), name="A", dtype=in_dtype)
2   B = te.placeholder((K, N), name="B", dtype=in_dtype)
3   k = te.reduce_axis((0, K), name="k")
4   C = te.compute(
5       (M, N),
6       lambda i, j: te.sum(A[i, k].astype(accum_dtype) *
7       B[k, j].astype(accum_dtype), axis=k),
8       name="C",
9   )
10  args = [A, B, C]
11  func = te.create_prim_func(args)
12  module = tvm.IRModule({"main": func})
13  module = ansor.autotune(module)
14  executable = tvm.compile(module)
15  A, B, C: torch.Tensor
16  executable(A, B, C)
```

**❸ Triton**

```
1   @triton.jit
2   def matmul_kernel(
3       a_ptr, b_ptr, c_ptr,
4       M, N, K,
5       stride_am, stride_ak,
6       stride_bk, stride_bn,
7       stride_cm, stride_cn,
8       BLOCK_SIZE_M: tl.constexpr, BLOCK_SIZE_N: tl.constexpr, BLOCK_SIZE_K: tl.constexpr,
9       GROUP_SIZE_M: tl.constexpr,
10      ACTIVATION: tl.constexpr
11  ):
12      pid = tl.program_id(axis=0)
13      num_pid_m = tl.cdiv(M, BLOCK_SIZE_M)
14      num_pid_n = tl.cdiv(N, BLOCK_SIZE_N)
15      num_pid_in_group = GROUP_SIZE_M * num_pid_n
16      group_id = pid // num_pid_in_group
17      first_pid_m = group_id * GROUP_SIZE_M
18      group_size_m = min(num_pid_m - first_pid_m, GROUP_SIZE_M)
19      pid_m = first_pid_m + ((pid % num_pid_in_group) % group_size_m)
20      pid_n = (pid % num_pid_in_group) // group_size_m
21      offs_am = (pid_m * BLOCK_SIZE_M + tl.arange(0, BLOCK_SIZE_M)) % M
22      offs_bn = (pid_n * BLOCK_SIZE_N + tl.arange(0, BLOCK_SIZE_N)) % N
23      offs_k = tl.arange(0, BLOCK_SIZE_K)
24      a_ptrs = a_ptr + (offs_am[:, None] * stride_am + offs_k[None, :] * stride_ak)
25      b_ptrs = b_ptr + (offs_k[:, None] * stride_bk + offs_bn[None, :] * stride_bn)
26      accumulator = tl.zeros((BLOCK_SIZE_M, BLOCK_SIZE_N), dtype=tl.float32)
27      for k in range(0, tl.cdiv(K, BLOCK_SIZE_K)):
28          a = tl.load(a_ptrs, mask=offs_k[None, :] < K - k * BLOCK_SIZE_K, other=0.0)
29          b = tl.load(b_ptrs, mask=offs_k[:, None] < K - k * BLOCK_SIZE_K, other=0.0)
30          accumulator = tl.dot(a, b, accumulator)
31          a_ptrs += BLOCK_SIZE_K * stride_ak
32          b_ptrs += BLOCK_SIZE_K * stride_bk
33      c = accumulator.to(tl.float16)
34      offs_cm = pid_m * BLOCK_SIZE_M + tl.arange(0, BLOCK_SIZE_M)
35      offs_cn = pid_n * BLOCK_SIZE_N + tl.arange(0, BLOCK_SIZE_N)
36      c_ptrs = c_ptr + stride_cm * offs_cm[:, None] + stride_cn * offs_cn[None, :]
37      c_mask = (offs_cm[:, None] < M) & (offs_cn[None, :] < N)
38      tl.store(c_ptrs, c, mask=c_mask)
39
40  def matmul(a, b):
41      ...
42      grid = lambda META: (cdiv(M, META['BLOCK_SIZE_M']) * cdiv(N, META['BLOCK_SIZE_N']))
43      matmul_kernel[grid](
44          a, b, c,
45          M, N, K,
46          a.stride(0), a.stride(1),
47          b.stride(0), b.stride(1),
48          c.stride(0), c.stride(1),
49          ACTIVATION=activation
50      )
51      return c
```

Figure 12: Side-by-side diff of minimal GEMM implementations in TVM, TILELANG, and Triton (first page).

```
1   @T.prim_func
2   def Matmul(
3       A: T.Tensor((M, K), "float16"),
4       B: T.Tensor((K, N), "float16"),
5       C: T.Tensor((M, N), "float16"),
6   ):
7       with T.Kernel(
8           ceildiv(N, block_N),
9           ceildiv(M, block_M), threads=threads) as (bx, by):
10
11          # Buffer Allocation
12          A_shared = T.alloc_shared((block_M, block_K), "float16")
13
14          B_shared = T.alloc_shared((block_K, block_N), "float16")
15
16          C_local = T.alloc_fragment((block_M, block_N), "float16")
17
18          # Initialize C_local
19          T.clear(C_local)
20
21          # Main Loop with Expanded Pipeline
22          T.copy(A[by * block_M, 0 * block_K], A_shared)
23
24          T.copy(B[0 * block_K, bx * block_N], B_shared)
25
26          # Main Loop with Pipeline Annotation
27          for k in T.serial(ceildiv(K, block_K)):
28              T.gemm(A_shared, B_shared, C_local)
29
30              T.copy(A[by * block_M, k * block_K], A_shared)
31
32              T.copy(B[k * block_K, bx * block_N], B_shared)
33
34          # Compute the last stage
35          T.gemm(A_shared, B_shared, C_local)
36          # Copy the result to the output buffer
37          T.copy(C_local, C[by * block_M, bx * block_N])
```

(a) Fused Tile Graph IR

```
1   @T.prim_func
2   def Matmul(
3       A: T.Tensor((M, K), "float16"), B: T.Tensor((K, N), "float16"),
4       C: T.Tensor((M, N), "float16"),
5   ):
6       A_shared = T.decl_buffer((block_M * block_K,), dtype, scope="shared")
7       B_shared = T.decl_buffer((block_K * block_N,), dtype, scope="shared")
8       C_local = T.decl_buffer((128,), accum_dtype, scope="local")
9       bx = T.thread_binding(T.ceildiv(N, block_N), "blockIdx.x")
10      by = T.thread_binding(T.ceildiv(M, block_M), "blockIdx.y")
11      tid = T.thread_binding(threads, "threadIdx.x")
12      for i in T.unroll(128):
13          C_local[i] = T.float32(0)
14      for i in T.unroll(4):
15          T.ptx_cp_async("uint8", A_shared.data, 0, A.data, 0, 16)
16          T.ptx_cp_async("uint8", B_shared.data, 0, B.data, 0, 16)
17      T.ptx_commit_group()
18      for ko in T.serial(ceildiv(K, block_K) - 1):
19          T.ptx_wait_group(0)
20          T.call_extern(
21              "tl::gemm_ss<128, 128, 32, 2, 2, 0, 0>",
22              T.tvm_access_ptr(A_shared.data, 0, block_M * block_K, 1),
23              T.tvm_access_ptr(B_shared.data, 0, block_K * block_N, 1),
24              T.tvm_access_ptr(C_local.data, 0, 128, 3),
25          )
26          for i in T.unroll(4):
27              T.ptx_cp_async("uint8", A_shared.data, 0, A.data, 0, 16)
28              T.ptx_cp_async("uint8", B_shared.data, 0, B.data, 0, 16)
29          T.ptx_commit_group()
30      T.ptx_wait_group(0)
31      T.call_extern(
32          "tl::gemm_ss<128, 128, 32, 2, 2, 0, 0>",
33          T.tvm_access_ptr(A_shared.data, 0, block_M * block_K, 1),
34          T.tvm_access_ptr(B_shared.data, 0, block_K * block_N, 1),
35          T.tvm_access_ptr(C_local.data, 0, 128, 3),
36      )
37      ...
```

(b) Lowered Tensor IR

Figure 13: A side-by-side comparison showing how tile-level FTG-IR is lowered into Tensor IR.

# K Comparison of FlashMLA Implementations on Different Architectures: NVIDIA vs. AMD

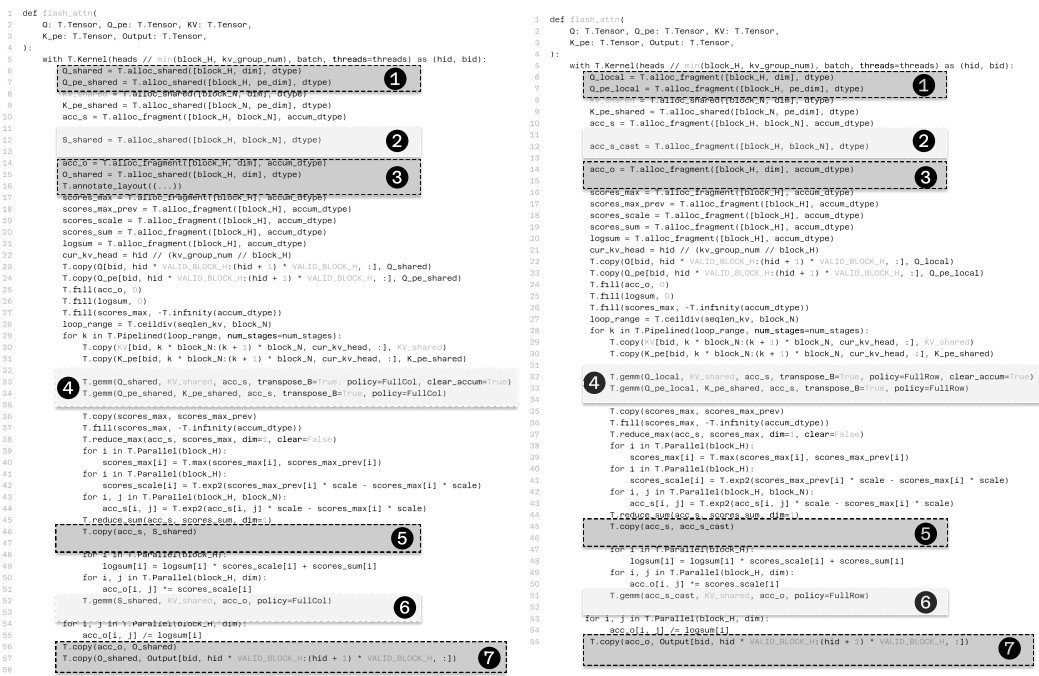

Figure 14: Comparison of FlashMLA implementations targeting NVIDIA (left) and AMD (right) architectures.

Figure 14 illustrates the code-level divergences between the FlashMLA implementations for NVIDIA and AMD architectures. While the high-level algorithmic structure remains unified, the implementation diverges to exploit distinct architectural strengths. First, Region 1 highlights the memory scope allocation for Query ($Q$) tiles, where the NVIDIA backend utilizes shared memory while the AMD backend prioritizes register memory. Similarly, Regions 2 and 5 collectively illustrate the divergence in intermediate accumulator storage: NVIDIA buffers these results in shared memory, whereas AMD maintains them directly in registers. Regarding output strategy, Regions 3 and 7 show that NVIDIA stages the output tile in shared memory to ensure coalesced transactions, in contrast to AMD, which performs a direct copy from registers to global memory. Finally, Regions 4 and 6 depict the adaptation of GEMM policies, employing a FullCol strategy for NVIDIA and FullRow for AMD to ensure optimal instruction performance.

## L Comparison With Recent Systems

Table 7 evaluates Causal MHA on an H100 ($B = 64, H = 64, D = 128$). TileLang is more concise than Tilus (Ding et al., 2025) (66 vs. 83 LOC) while achieving a 59%-75% performance gain. Table 8 evaluates the same MHA on an RTX 4090 ($B = 16, H = 32, D = 128$). TileLang achieves the best performance among baselines and outperforms the latest Tilus by 3%–4% with significantly fewer lines of code.

| seq_len | Tilus (ms) | TileLang (ms) |
|---------|-----------|---------------|
| 1024 | 6.85 | 4.29 |
| 2048 | 25.65 | 15.69 |
| 4096 | 97.14 | 55.44 |

Table 7: Performance of Causal MHA on H100 ($B = 64, H = 64, D = 128$).

|  | seq_len | FA2 (tflops) | Triton (tflops) | Tilus (tflops) | TileLang (tflops) |
|---|---|---|---|---|---|
| MHA | 1024 | 137.58 | 106.61 | 133.82 | 143.40 |
|  | 2048 | 151.46 | 121.44 | 154.36 | 159.03 |
|  | 4096 | 162.70 | 129.75 | 159.26 | 162.79 |
|  | 8192 | 164.10 | 134.40 | 161.60 | 165.56 |
| LoC |  | 389 | 197 | 393 | 138 |

Table 8: Performance and Lines of Code(LoC) of Causal MHA on RTX 4090 ($B = 16, H = 32, D = 128$).

| $(M, N, K)$ | TileLang (ms) | Gluon (ms) | Helion (ms) | Tilus (ms) |
|---|---|---|---|---|
| 8192, 1024, 8192 | 0.16 | 0.30 | 0.26 | 0.31 |
| 8192, 8192, 8192 | 1.40 | 2.22 | 1.64 | 2.97 |
| 8192, 28672, 8192 | 5.09 | 7.82 | 5.87 | 10.19 |
| 8192, 8192, 28672 | 5.09 | 7.77 | 5.89 | 10.02 |

Table 9: Comparison of TileLang, Gluon (LoC = 68), Helion (LoC = 24), and Tilus (LoC = 110) on GEMM workloads.

| seq_len | TileLang (ms) | Helion (ms) |
|---|---|---|
| 1024 | 0.17 | 0.19 |
| 2048 | 0.33 | 0.36 |
| 4096 | 0.65 | 0.71 |
| 8192 | 1.28 | 1.41 |
| 16384 | 2.53 | 3.01 |
| 32768 | 5.08 | 5.56 |

Table 10: Performance of Mamba-chunk-scan on H100, with batch size 8, 80 attention heads, model dimension 64, dstate 128, and sequence lengths ranging from 1024 to 32768. Helion LOC=116, TileLang LOC=114.

We evaluate the latest official Helion (PyTorch, 2025) and Gluon (OpenAI, 2025) examples that support execution on Hopper, covering both GEMM and Mamba-chunk-scan workloads. As shown in Table 9, TileLang achieves $1.15$–$1.62\times$, $1.52$–$1.83\times$, and $1.87$–$2.12\times$ speedups over Helion, Gluon, and Tilus, respectively.

On Mamba-2-chunk-scan, TileLang further provides $1.10$–$1.19\times$ speedups over Helion, as reported in Table 10.

The differences are attributable to design limitations in existing DSLs. Helion lacks an effective tile-recommendation system, making optimization difficult and causing long tuning times (over 20 minutes). Gluon lacks appropriate abstractions and interfaces for pipeline scheduling, which makes it difficult to achieve effective overlap of computation and memory operations. Second, Gluon operates at a lower programming abstraction level, requiring users to manually make a larger number of design and optimization decisions. As a result, writing high-performance kernels in Gluon is considerably more challenging. This is also reflected by the fact that the Gluon implementation of GEMM requires substantially more lines of code than the corresponding TileLang implementation. Tilus exposes thread-block-level control over shared memory and registers, but without tile abstraction and critical tile-related optimization. In contrast, TileLang's tile-recommendation mechanism efficiently identifies good tile configurations, and its pipeline-inference strategy generates an effective schedule that overlaps computation and memory I/O. These additions provide a clearer qualitative and quantitative comparison among recent DSLs and further highlight TILELANG's performance and usability advantages.

## M    BASELINE COMMIT HASHES

To ensure reproducibility, we provide the specific commit hashes for the baselines used in our experiments that were not specified in the main text (see Table 11).

Table 11: Commit hashes for baseline frameworks.

| System | Commit Hash |
|---|---|
| Gluon | 61cef5bdbfe2f179208f057d72c0b43b4885e5d2 |
| Helion | 9a30bd18dcd87e08784691d5799e1af71fe0502f |
| Tilus | 505f566210319e2f55eeeddc393f01a203950510 |
| Ansor | 64969035fd4f3c1ddcc23caa84567bf90e33889c |
| ThunderKittens | 572073a3935f91a268d37d5262cee0d950c2e9b2 |
| Marlin | 1f25790bdd49fba53106164a24666dade68d7c90 |
| Block-Sparse-Attention | 6ec5a27a0cd6bd92ea6296698d64e460c73da27e |
| ComposableKernel | b8893b933963e86b76fa3fa088ededc4504119f9 |
| Tilus (Variant?) | b2085f5ea08c504efeb2cab7cfaae3cd99701634 |
| FlashAttention-2 (FA2) | 5d2cd3bcbaeff6fe1bfc5d0ff489451b0d4827a6 |
| TritonBench | a490be73ba84ab977de5cf78055a1dcb2e314f40 |

## N    OPERATOR SHAPES IN OUR BENCHMARK

Table 12: Matrix shapes in our FP16 Matmul evaluation

|   | D0 | D1 | D2 | D3 |
|---|---|---|---|---|
| M | 8192 | 8192 | 8192 | 8192 |
| N | 1024 | 8192 | 28672 | 8192 |
| K | 8192 | 8192 | 8192 | 28672 |

Table 13: Matrix shapes in our Fused Dequantize-Matmul evaluation

|   | M0 | M1 | M2 | M3 |
|---|---|---|---|---|
| M | 1 | 1 | 1 | 1 |
| N | 1024 | 8192 | 28672 | 8192 |
| K | 8192 | 8192 | 8192 | 28672 |

Table 14: FlashAttention and Block Sparse Attention(with 50%, 90% sparsity) shapes in our evaluation

|   | FA0 | FA1 | FA2 | FA3 | FA4 | FA5 | FA6 | FA7 |
|---|---|---|---|---|---|---|---|---|
| batch | 64 | 64 | 64 | 64 | 64 | 64 | 64 | 64 |
| nheads | 64 | 64 | 64 | 64 | 64 | 64 | 64 | 64 |
| seq_len | 1024 | 2048 | 4096 | 8192 | 1024 | 2048 | 4096 | 8192 |
| head_dim | 128 | 128 | 128 | 128 | 128 | 128 | 128 | 128 |
| causal | false | false | false | false | true | true | true | true |

Table 15: FlashMLA shapes in our evaluation

|   | FMLA0 | FMLA1 | FMLA2 | FMLA3 |
|---|---|---|---|---|
| batch | 64 | 64 | 64 | 64 |
| nheads | 128 | 128 | 128 | 128 |
| seq_len | 1024 | 2048 | 4096 | 8192 |
| head_dim | 512 | 512 | 512 | 512 |
| pe_dim | 64 | 64 | 64 | 64 |
| causal | false | false | false | false |

Table 16: Convolution-2D shapes in our evaluation

|   | Conv0 | Conv1 | Conv2 | Conv3 | Conv4 | Conv5 | Conv6 | Conv7 |
|---|---|---|---|---|---|---|---|---|
| N | 128 | 128 | 128 | 128 | 128 | 128 | 128 | 128 |
| C | 2048 | 512 | 512 | 512 | 256 | 1024 | 512 | 64 |
| H | 7 | 7 | 14 | 7 | 14 | 14 | 28 | 56 |
| W | 7 | 7 | 14 | 7 | 14 | 14 | 28 | 56 |
| F | 512 | 2048 | 512 | 512 | 256 | 256 | 128 | 64 |
| K | 1 | 1 | 3 | 3 | 3 | 1 | 1 | 1 |
| S | 1 | 1 | 2 | 1 | 1 | 1 | 1 | 1 |
| D | 1 | 1 | 1 | 1 | 1 | 1 | 1 | 1 |
| P | 0 | 0 | 1 | 1 | 1 | 0 | 0 | 0 |
| G | 1 | 1 | 1 | 1 | 1 | 1 | 1 | 1 |
| HO | 7 | 7 | 7 | 7 | 14 | 14 | 28 | 56 |
| WO | 7 | 7 | 7 | 7 | 14 | 14 | 28 | 56 |
| Count | 2 | 3 | 1 | 2 | 5 | 5 | 3 | 1 |

Table 17: Chunk-Gated-Delta-Net kernel shapes in our evaluation

|          | CGDN0 | CGDN1 | CGDN2 | CGDN3 | CGDN4 | CGDN5 |
|----------|-------|-------|-------|-------|-------|-------|
| batch    | 1     | 1     | 1     | 64    | 64    | 64    |
| nheads   | 32    | 32    | 32    | 32    | 32    | 32    |
| seq_len  | 16384 | 32768 | 65536 | 1024  | 2048  | 4096  |
| head_dim | 128   | 128   | 128   | 128   | 128   | 128   |

Table 18: Vertical Slash Sparse Attention shapes in our evaluation

|               | VSSA0 | VSSA1 | VSSA2 | VSSA3 |
|---------------|-------|-------|-------|-------|
| batch         | 1     | 1     | 1     | 1     |
| nheads        | 1     | 1     | 1     | 1     |
| seq_len       | 8192  | 16384 | 32768 | 65536 |
| head_dim      | 64    | 64    | 64    | 64    |
| vertical size | 1000  | 1000  | 800   | 1000  |
| slash size    | 600   | 200   | 600   | 600   |

Table 19: Attention Sink shapes in our evaluation

|            | Sink0 | Sink1 | Sink2 | Sink3 |
|------------|-------|-------|-------|-------|
| batch      | 1     | 1     | 1     | 1     |
| nheads     | 64    | 64    | 64    | 64    |
| kv_heads   | 8     | 8     | 8     | 8     |
| seq_len    | 1024  | 2048  | 4096  | 8192  |
| kv_seq_len | 1024  | 2048  | 4096  | 8192  |
| head_dim   | 64    | 64    | 64    | 64    |
| casual     | true  | true  | true  | true  |

## O  KERNEL IMPLEMENTATIONS

### O.1  MATRIX MULTIPLICATION (MATMUL)

```python
@tilelang.jit
def Matmul(A: T.Tensor, B: T.Tensor, C: T.Tensor):
    with T.Kernel(N // block_N, M // block_M,
        threads=threads) as (bx, by):
        A_shared = T.alloc_shared(block_M, block_K)
        B_shared = T.alloc_shared(block_K, block_N)
        C_local = T.alloc_fragment(block_M, block_N)

        T.clear(C_local)
        for k in T.Pipelined(K // block_K, num_stages=2):
            T.copy(A[by * block_M, k * block_K], A_shared)
            T.copy(B[k * block_K, bx * block_N], B_shared)
            T.gemm(A_shared, B_shared, C_local)

        T.copy(C_local, C[by * block_M, bx * block_N])
```

Figure 15: Kernel Implementation of Matrix Multiplication.

### O.2  DEQUANTIZED MATRIX MULTIPLICATION

```python
@tilelang.jit
def dequantize_gemv(A: T.Tensor, B: T.Tensor, C: T.Tensor):
    with T.Kernel(T.ceildiv(N, n_partition), M, threads=(reduce_thread, n_partition)) as (bx, by):
        A_local = T.alloc_local((micro_size_k,), in_dtype)
        B_quant_local = T.alloc_local([micro_size_k_compressed], storage_dtype)
        B_dequantize_local = T.alloc_local([micro_size_k], in_dtype)
        accum_res = T.alloc_local((1,), accum_dtype)
        reduced_accum_res = T.alloc_local((1,), accum_dtype)

        T.clear(accum_res)
        for ko in T.serial(T.ceildiv(K, block_K)):
            for v in T.vectorized(micro_size_k):
                A_local[v] = A[by, ko * block_K + kr * micro_size_k + v]

            for v in T.vectorized(micro_size_k_compressed):
                B_quant_local[v] = B[
                    bx * n_partition + ni,
                    ko * (reduce_thread * micro_size_k_compressed) +
                    kr * micro_size_k_compressed + v,
                ]

            T.call_extern(
                "fast_decode_int4",
                T.address_of(B_quant_local[0]),
                T.address_of(B_dequantize_local[0]),
                dtype=in_dtype,
            )

            for ki in T.serial(micro_size_k):
                accum_res[0] += A_local[ki] * B_dequantize_local[ki]

        with T.attr(
                T.comm_reducer(lambda x, y: x + y, [T.Cast(accum_dtype, 0)]),
                "reduce_scope",
                T.reinterpret(T.uint64(0), dtype="handle"),
        ):
            T.evaluate(
                T.tvm_thread_allreduce(
                    T.uint32(1),
                    accum_res[0],
                    True,
                    reduced_accum_res[0],
                    kr,
                    dtype="handle",
                ))
        if kr == 0:
            C[by, bx * n_partition + ni] = reduced_accum_res[0]
```

Figure 16: Implementation of Weight-Only Quantization ($W_{\text{FP4\_E2M1}}A_{\text{FP16}}$) Matmul using TILE-LANG, showcasing support for mixed-precision computations via a simple form.

## O.3 FLASH ATTENTION IMPLEMENTATION

```python
@tilelang.jit
def flash_attention(Q: T.Tensor, K: T.Tensor, V: T.Tensor, Output: T.Tensor):
    with T.Kernel(
            T.ceildiv(seq_len, block_M), heads, batch, threads=threads) as (bx, by, bz):
        Q_shared = T.alloc_shared([block_M, dim], dtype)
        K_shared = T.alloc_shared([block_N, dim], dtype)
        V_shared = T.alloc_shared([block_N, dim], dtype)
        O_shared = T.alloc_shared([block_M, dim], dtype)
        acc_s = T.alloc_fragment([block_M, block_N], accum_dtype)
        acc_s_cast = T.alloc_fragment([block_M, block_N], dtype)
        acc_o = T.alloc_fragment([block_M, dim], accum_dtype)
        scores_max = T.alloc_fragment([block_M], accum_dtype)
        scores_max_prev = T.alloc_fragment([block_M], accum_dtype)
        scores_scale = T.alloc_fragment([block_M], accum_dtype)
        scores_sum = T.alloc_fragment([block_M], accum_dtype)
        logsum = T.alloc_fragment([block_M], accum_dtype)

        T.copy(Q[bz, bx * block_M:(bx + 1) * block_M, by, :], Q_shared)
        T.fill(acc_o, 0)
        T.fill(logsum, 0)
        T.fill(scores_max, -T.infinity(accum_dtype))

        loop_range = (
            T.min(T.ceildiv(seq_len, block_N), T.ceildiv(
                (bx + 1) * block_M, block_N)) if is_causal else T.ceildiv(seq_len, block_N))

        for k in T.Pipelined(loop_range, num_stages=num_stages):
            T.copy(K[bz, k * block_N:(k + 1) * block_N, by, :], K_shared)
            if is_causal:
                for i, j in T.Parallel(block_M, block_N):
                    acc_s[i, j] = T.if_then_else(bx * block_M + i >= k * block_N + j, 0,
                                                 -T.infinity(acc_s.dtype))
            else:
                T.clear(acc_s)
            T.gemm(Q_shared, K_shared, acc_s, transpose_B=True, policy=T.GemmWarpPolicy.FullRow)
            T.copy(scores_max, scores_max_prev)
            T.fill(scores_max, -T.infinity(accum_dtype))
            T.reduce_max(acc_s, scores_max, dim=1, clear=False)
            for i in T.Parallel(block_M):
                scores_scale[i] = T.exp2(scores_max_prev[i] * scale - scores_max[i] * scale)
            for i, j in T.Parallel(block_M, block_N):
                acc_s[i, j] = T.exp2(acc_s[i, j] * scale - scores_max[i] * scale)
            T.reduce_sum(acc_s, scores_sum, dim=1)
            for i in T.Parallel(block_M):
                logsum[i] = logsum[i] * scores_scale[i] + scores_sum[i]
            T.copy(acc_s, acc_s_cast)
            for i, j in T.Parallel(block_M, dim):
                acc_o[i, j] *= scores_scale[i]
            T.copy(V[bz, k * block_N:(k + 1) * block_N, by, :], V_shared)
            T.gemm(acc_s_cast, V_shared, acc_o, policy=T.GemmWarpPolicy.FullRow)
        for i, j in T.Parallel(block_M, dim):
            acc_o[i, j] /= logsum[i]
        T.copy(acc_o, O_shared)
        T.copy(O_shared, Output[bz, bx * block_M:(bx + 1) * block_M, by, :])
```

Figure 17: Implementation of Flash Attention with TILELANG.

## O.4 FLASHMLA IMPLEMENTATION

```
1   @tilelang.jit
2   def flash_mla(
3           Q: T.Tensor([batch, heads, dim], dtype),
4           Q_pe: T.Tensor([batch, heads, pe_dim], dtype),
5           KV: T.Tensor([batch, seqlen_kv, kv_head_num, dim], dtype),
6           K_pe: T.Tensor([batch, seqlen_kv, kv_head_num, pe_dim], dtype),
7           Output: T.Tensor([batch, heads, dim], dtype),
8   ):
9       with T.Kernel(batch, heads // min(block_H, kv_group_num), threads=256) as (bx, by):
10          Q_shared = T.alloc_shared([block_H, dim], dtype)
11          S_shared = T.alloc_shared([block_H, block_N], dtype)
12          Q_pe_shared = T.alloc_shared([block_H, pe_dim], dtype)
13          KV_shared = T.alloc_shared([block_N, dim], dtype)
14          K_pe_shared = T.alloc_shared([block_N, pe_dim], dtype)
15          O_shared = T.alloc_shared([block_H, dim], dtype)
16          acc_s = T.alloc_fragment([block_H, block_N], accum_dtype)
17          acc_o = T.alloc_fragment([block_H, dim], accum_dtype)
18          scores_max = T.alloc_fragment([block_H], accum_dtype)
19          scores_max_prev = T.alloc_fragment([block_H], accum_dtype)
20          scores_scale = T.alloc_fragment([block_H], accum_dtype)
21          scores_sum = T.alloc_fragment([block_H], accum_dtype)
22          logsum = T.alloc_fragment([block_H], accum_dtype)
23
24          cur_kv_head = by // (kv_group_num // block_H)
25          T.use_swizzle(10)
26
27          T.copy(Q[bx, by * VALID_BLOCK_H:(by + 1) * VALID_BLOCK_H, :], Q_shared)
28          T.copy(Q_pe[bx, by * VALID_BLOCK_H:(by + 1) * VALID_BLOCK_H, :], Q_pe_shared)
29          T.fill(acc_o, 0)
30          T.fill(logsum, 0)
31          T.fill(scores_max, -T.infinity(accum_dtype))
32
33          loop_range = T.ceildiv(seqlen_kv, block_N)
34          for k in T.Pipelined(loop_range, num_stages=2):
35              T.copy(KV[bx, k * block_N:(k + 1) * block_N, cur_kv_head, :], KV_shared)
36              T.copy(K_pe[bx, k * block_N:(k + 1) * block_N, cur_kv_head, :], K_pe_shared)
37              T.clear(acc_s)
38              T.gemm(
39                  Q_shared, KV_shared, acc_s, transpose_B=True, policy=T.GemmWarpPolicy.FullCol)
40              T.gemm(
41                  Q_pe_shared,
42                  K_pe_shared,
43                  acc_s,
44                  transpose_B=True,
45                  policy=T.GemmWarpPolicy.FullCol)
46              T.copy(scores_max, scores_max_prev)
47              T.fill(scores_max, -T.infinity(accum_dtype))
48              T.reduce_max(acc_s, scores_max, dim=1, clear=False)
49              for i in T.Parallel(block_H):
50                  scores_scale[i] = T.exp2(scores_max_prev[i] * scale - scores_max[i] * scale)
51              for i, j in T.Parallel(block_H, block_N):
52                  acc_s[i, j] = T.exp2(acc_s[i, j] * scale - scores_max[i] * scale)
53              T.reduce_sum(acc_s, scores_sum, dim=1)
54              T.copy(acc_s, S_shared)
55              for i in T.Parallel(block_H):
56                  logsum[i] = logsum[i] * scores_scale[i] + scores_sum[i]
57              for i, j in T.Parallel(block_H, dim):
58                  acc_o[i, j] *= scores_scale[i]
59              T.gemm(S_shared, KV_shared, acc_o, policy=T.GemmWarpPolicy.FullCol)
60          for i, j in T.Parallel(block_H, dim):
61              acc_o[i, j] /= logsum[i]
62          T.copy(acc_o, O_shared)
63          T.copy(O_shared, Output[bx, by * VALID_BLOCK_H:(by + 1) * VALID_BLOCK_H, :])
```

Figure 18: Implementation of FlashMLA with TILELANG.

## O.5 BLOCK SPARSE ATTENTION IMPLEMENTATION

```python
@tilelang.jit
def blocksparse_attn(Q: T.Tensor, K: T.Tensor, V: T.Tensor, BlockMask: T.Tensor, Output: T.Tensor):
    with T.Kernel(
            T.ceildiv(seq_len, block_M), heads, batch, threads=threads) as (bx, by, bz):
        Q_shared = T.alloc_shared([block_M, dim], dtype)
        K_shared = T.alloc_shared([block_N, dim], dtype)
        V_shared = T.alloc_shared([block_N, dim], dtype)
        O_shared = T.alloc_shared([block_M, dim], dtype)
        acc_s = T.alloc_fragment([block_M, block_N], accum_dtype)
        acc_s_cast = T.alloc_fragment([block_M, block_N], dtype)
        acc_o = T.alloc_fragment([block_M, dim], accum_dtype)
        scores_max = T.alloc_fragment([block_M], accum_dtype)
        scores_max_prev = T.alloc_fragment([block_M], accum_dtype)
        scores_scale = T.alloc_fragment([block_M], accum_dtype)
        scores_sum = T.alloc_fragment([block_M], accum_dtype)
        logsum = T.alloc_fragment([block_M], accum_dtype)

        T.copy(Q[bz, bx * block_M:(bx + 1) * block_M, by, :], Q_shared)
        T.fill(acc_o, 0)
        T.fill(logsum, 0)
        T.fill(scores_max, -T.infinity(accum_dtype))

        loop_range = (
            T.min(T.ceildiv(seq_len, block_N), T.ceildiv(
                (bx + 1) * block_M, block_N)) if is_causal else T.ceildiv(seq_len, block_N))

        for k in T.Pipelined(loop_range, num_stages=num_stages):
            if BlockMask[bz, bx, by, k]:
                T.copy(K[bz, k * block_N:(k + 1) * block_N, by, :], K_shared)
                if is_causal:
                    for i, j in T.Parallel(block_M, block_N):
                        acc_s[i, j] = T.if_then_else(bx * block_M + i >= k * block_N + j, 0,
                                                     -T.infinity(acc_s.dtype))
                else:
                    T.clear(acc_s)
                T.gemm(Q_shared, K_shared, acc_s, transpose_B=True, policy=T.GemmWarpPolicy.FullRow)
                T.copy(scores_max, scores_max_prev)
                T.fill(scores_max, -T.infinity(accum_dtype))
                T.reduce_max(acc_s, scores_max, dim=1, clear=False)
                for i in T.Parallel(block_M):
                    scores_scale[i] = T.exp2(scores_max_prev[i] * scale - scores_max[i] * scale)
                for i, j in T.Parallel(block_M, block_N):
                    acc_s[i, j] = T.exp2(acc_s[i, j] * scale - scores_max[i] * scale)
                T.reduce_sum(acc_s, scores_sum, dim=1)
                for i in T.Parallel(block_M):
                    logsum[i] = logsum[i] * scores_scale[i] + scores_sum[i]
                T.copy(acc_s, acc_s_cast)
                for i, j in T.Parallel(block_M, dim):
                    acc_o[i, j] *= scores_scale[i]
                T.copy(V[bz, k * block_N:(k + 1) * block_N, by, :], V_shared)
                T.gemm(acc_s_cast, V_shared, acc_o, policy=T.GemmWarpPolicy.FullRow)
        for i, j in T.Parallel(block_M, dim):
            acc_o[i, j] /= logsum[i]
        T.copy(acc_o, O_shared)
        T.copy(O_shared, Output[bz, bx * block_M:(bx + 1) * block_M, by, :])
```

Figure 19: Implementation of Block Sparse Flash Attention with TILELANG.

