# OpenReview forum: "TileLang: Bridge Programmability and Performance in Modern Neural Kernels"
_ICLR.cc/2026/Conference — ICLR 2026 Oral_

### Official Review · Reviewer_7JBZ · 2025-10-23

**Soundness:** 4
**Presentation:** 2
**Contribution:** 4
**Rating:** 8
**Confidence:** 4

**Summary:**

TileLang is a tile-level domain-specific language (DSL) for developing high-performance AI kernels. It is built on top of TVM, and provides {unified fused tile-level dataflow graph (FTG), tile recommendation, tile inference}, and explicit control over {memory placement, data movement, parallel scheduling} to bridge the gap between programmability and performance.

Overall, I like the paper.

**Strengths:**

**S1.** Excellent contribution from both research and engineering perspectives. TileLang addresses a real pain point in ML kernel development by bridging the programmability-performance gap, which will greatly help in migrating ML kernels to new hardware platforms, as well as developing new ML kernels on existing hardware.

**S2.** Strong empirical results across diverse kernels and hardware platforms. The benchmarks demonstrate substantial speedups (over Triton, PyTorch) and sufficiently close to the hand-crafted kernels (e.g. FlashMLA).

**S3.** The unified FTG abstraction combined with tile recommendation and tile inference provides a principled approach to kernel optimization. The two-stage workflow effectively balances automation with fine-grained developer control.

**Weaknesses:**

(Here, the weaknesses are more about how the paper writing can be improved to provide more informative and intuitive insights to the readers, but less about the technical contributions.)

**W1.** Missing comparison with TVM. Since TileLang is built directly on top of TVM, the paper lacks benchmarks comparing against TVM itself. While comparisons with Triton/Torch/hand-crafted kernels are valuable, without TVM we're missing a key component to understand TileLang's contribution over its foundation.

**W2.** Lack of software architecture overview. The relationship between TileLang and TVM is unclear. I would like to see an illustration showing the compilation pipeline: Python TileLang DSL → FTG → TVM IR (TensorIR/Relay IR?) → CUDA. What is the relationship between FTG and TVM's IRs? Is FTG a standalone IR or must it combine with other IRs to fully represent the computation? This is not stated clearly in the paper and would help distinguish TileLang's real contribution from TVM's existing capabilities.

**W3.** Missing direct code comparisons with TVM/Triton/etc. Instead of just describing syntactic differences and abstraction levels, showing actual side-by-side Python code comparing TileLang vs TVM vs Triton would be much more intuitive. ("Shut up and show me the code").

**W4.** No comparison with TaichiLang, which sits at a higher abstraction level and supports autodiff. A performance/LoC/syntax comparison would help position TileLang in the landscape of kernel programming frameworks. Or, at least, a mention of the differences.

**Questions:**

**Q1.** I was wondering about the benchmarking details - do your performance numbers include the Python wrapper overhead, or are they just measuring the kernel execution time? Also, what warmup strategy did you use and how many repetitions for the statistics?

**Q2.** It would be helpful to see some measurements of compilation and code generation time in your benchmarks. This would help understand the overhead for first-time graph execution, especially for JIT compilation scenarios.

**Q3.** Similar to W3 asking for direct code comparisons, could you provide a direct IR-level comparison between TileLang and TVM for a simple matmul example? It would be very helpful to see the actual IR representations side-by-side to understand how different TileLang's tile-level abstractions are from TVM's TensorIR at the IR level.

---

> ### Author Response · Authors · 2025-11-20
> **Response to Reviewer 7JBZ**
>
> We sincerely thank the reviewer for the thoughtful and detailed comments. The feedback has been invaluable in improving the clarity, completeness, and overall presentation of the paper. We carefully addressed each point raised and believe the revisions significantly strengthen the work.
>
> ## W1. Missing comparison with TVM
>
> We thank the reviewer for raising this point. As suggested, we now include a direct comparison with TVM. We evaluate TileLang against Ansor, TVM’s auto-scheduler, on the GEMM kernel on H100. TileLang achieves much shorter tuning time and delivers substantially higher FLOPs, while Ansor’s ML-based search incurs long tuning times and cannot generate Tensor Core kernels on modern GPUs. TileLang’s advantages come from its first-class tile IR and hardware-aware cost model, which greatly reduce the search space and enable high-performance configurations.
>
> These results clarify TileLang’s contribution beyond TVM and highlight the benefits of TileLang’s tile-centric design.
>
> ### GEMM @ H100
>
> | Config (M,N,K) | TileLang tuning time (s, 20 cfgs) | TileLang perf (TFLOPs) | TVM Ansor tuning time (s, 100 trials) | TVM Ansor perf (TFLOPs) |
> | --- | ---: | ---: | ---: | ---: |
> | 8192,1024,8192 | 11.81 | 613.74 | 518.52 | 22.76 |
> | 8192,8192,8192 | 11.78 | 661.17 | 455.51 | 26.89 |
> | 8192,28672,8192 | 14.59 | 676.17 | 3007.00 | 26.31 |
> | 8192,8192,28672 | 14.31 | 680.32 | 4142.05 | 24.23 |
>
> ## W2. Lack of software architecture overview
>
> What is the relationship between FTG and TVM’s IRs? Is FTG a standalone IR or must it combine with other IRs to fully represent the computation?
>
> We appreciate the reviewer’s question. As suggested, we have added a clear software-architecture overview in Section 5 in the revised paper to describe the full compilation pipeline: TileLang DSL → FTG → TensorIR → CUDA/ROCm. TileLang is built on top of the TVM backend, but its main contributions sit above TVM. TileLang introduces a tile-level Python DSL, the fused tile-level dataflow IR (FTG), and a hardware-aware recommendation and inference pipeline. These components define TileLang’s programming and optimization model and are not provided by TVM.
>
> In practice, TileLang programs are written using compiler-visible tile primitives (e.g., `T.alloc`, `T.copy`, `T.gemm`). The DSL first lowers to FTG, which is a tile-level IR that captures the computation in terms of tile shapes, swizzles, pipeline depth, and partitioning decisions. FTG represents the full tile dataflow and scheduling structure. After FTG is legalized, it is then lowered into TVM’s TensorIR, which provides the low-level loop, memory, and code-generation infrastructure. In other words, FTG defines the tile-level semantics and optimization space, while TensorIR serves as the backend to materialize these decisions into CUDA/ROCm kernels.
>
> In summary, TileLang provides the tile abstraction, FTG IR, and optimization passes, while TVM supplies the low-level backend.
>
> ## W3. Missing direct code comparisons with TVM/Triton/etc.
>
> We thank the reviewer for this valuable suggestion. Following the recommendation, we have added an Appendix I in the revised paper with actual side-by-side Python and IR code comparisons among TileLang, TVM, and Triton for a simple matrix-multiplication kernel. The comparison shows that TileLang’s tile-centric abstraction treats tiles as first-class IR constructs, greatly reducing code size, avoiding manual pointer arithmetic commonly required in Triton, and removing boilerplate schedule/compute separation in TVM. It also illustrates how TileLang offers clearer and more expressive control for complex operators. We refer the reviewer to Appendix I for the full code listings.
>
> ## W4. No comparison with TaichiLang
>
> We thank the reviewer for highlighting the missing comparison with TaichiLang. Although both Taichi and TileLang target GPU workloads, they differ fundamentally in abstraction and intended use. Taichi provides a high-level, scalar-loop DSL with automatic parallelization, SNode-based data layouts, and strong autodiff support, making it well suited for scientific computing and simulation.
>
> TileLang, by contrast, is designed for deep-learning kernels such as attention and GEMM, where peak performance requires explicit control of tile shapes, memory placement across shared/LDS and registers, multi-stage pipelining, warp specialization, and instruction selection (Tensor Core MMA or AMD MFMA). These capabilities are not directly expressible in Taichi’s fully automatic model.
>
> We now describe these differences in Related Work, clarifying TileLang’s position among GPU programming frameworks.

---

> ### Author Response · Authors · 2025-11-20
> **Response to Reviewer 7JBZ**
>
> ## Q1. Benchmarking details
>
> We thank the reviewer for asking about the benchmarking methodology.
>
> - Python-side overhead: Our reported numbers are operator-level wall-clock latencies measured from the Python call boundary to completion with an explicit device synchronization. They therefore include lightweight host overhead such as kernel launch and tensor-metadata checks, and reflect steady-state hot-cache execution. Inputs and outputs are pre-allocated on device, and we exclude H2D/D2H transfer and one-time JIT/compilation costs.
>
> - Warmup and repetitions: We trigger compilation with a dummy call, then discard 50 warmup iterations to stabilize caches and autotuned parameters. For measurement, we run 100 timed repetitions, synchronizing the default stream before and after each iteration (`torch.cuda.synchronize` on CUDA/ROCm), and report the median (mean ± std available upon request).
>
> ## Q2. Compilation and code generation time
>
> We thank the reviewer for this suggestion. We have added measurements of compilation and code-generation time to clarify the overhead for first-time execution under JIT scenarios. TileLang uses a multi-level cache consisting of an in-process JIT kernel cache and a disk cache containing compiled binaries. On a disk-cache miss, we specialize and compile the kernel once, and subsequently populate both caches.
>
> ### Compile times: GEMM (H100)
>
> - First compile: 3.20–5.76 s (avg 3.87 s)
> - Disk-cache hit: 19.9–23.4 ms (avg 21.5 ms)
> - In-process JIT cache hit: 0.74–0.76 µs (avg 0.75 µs)
>
> ### Compile times: FlashMHA (H100)
>
> - First compile: 3.97–6.62 s (avg 4.43 s)
> - Disk-cache hit: 62–76 ms (avg 66.6 ms)
> - In-process JIT cache hit: 1.17–1.27 µs (avg 1.19 µs)
>
> These results show that first-time compilation is amortized after the initial specialization, and subsequent executions benefit from extremely low cache-hit overhead.
>
> ## Q3. Direct IR-level comparison between TileLang and TVM for a simple matmul example
>
> We thank the reviewer for this helpful suggestion. As noted in our response to W2, we have clarified the relationship between FTG and TVM’s IRs. Following the reviewer’s suggestion, we have added a new Appendix J, which now provides a direct IR-level side-by-side comparison between TileLang’s FTG IR and TVM’s TensorIR for a simple matrix-multiplication example. This comparison illustrates how TileLang’s tile-level abstraction differs from low-level TVM’s loop-based IR.

---

> > ### Comment · Reviewer_7JBZ · 2025-11-20
> >
> > Thanks to the author for the response.
> >
> > W1/W2/W3/Q1/Q2/Q3 have been addressed.
> >
> > We still need to update the paper for W4. I don't see TaichiLang mentioned in the related work or citation in the updated paper PDF.
> >
> > Otherwise, LGTM! Looking forward to industrializing TileLang to support new and emerging hardware.

---

> > > ### Author Response · Authors · 2025-11-21
> > > **Response to Reviewer 7JBZ**
> > >
> > > ### C1. Need to update the paper for W4
> > >
> > > Thank you for the further comment. We have added TaichiLang in the related work section (Sec. 2) and included additional discussion in Appendix A (titled “Extended Discussion of Related Work”) to clarify its relationship to TileLang.
> > >
> > > [1] Y. Hu et al. Taichi: A Language for High-Performance Computation on Sparse Data Structures. SIGGRAPH Asia, 2019.
> > >
> > > [2] Y. Hu et al. DiffTaichi: Differentiable Programming for Physical Simulation. ICLR, 2020.
> > >
> > > [3] Y. Hu et al. QuanTaichi: A Compiler for Quantized Simulations. SIGGRAPH, 2021.
> > >
> > > We sincerely appreciate the reviewer’s encouragement and will continue advancing TileLang toward broader hardware support.

---

### Official Review · Reviewer_CKBd · 2025-10-24

**Soundness:** 3
**Presentation:** 3
**Contribution:** 4
**Rating:** 8
**Confidence:** 5

**Summary:**

The authors propose a new domain-specific language (DSL) for developing efficient kernels on modern GPUs, called TileLang. Similar to the widely used tile-level kernel programming language Triton, TileLang also treats tiles as first-class citizens. However, unlike Triton, TileLang exposes more control over underlying hardware components—such as memory scope, tile layout, and thread mapping—to the programmer. This additional control allows developers to implement more efficient kernels and achieve better performance in cases where the Triton compiler cannot automatically apply these optimizations. Extensive experiments demonstrate the advantages of TileLang over existing languages.

**Strengths:**

1. The problem addressed by this work is highly important. We need effective DSLs to simplify writing peak-performance kernels for modern GPUs, which are increasingly complex to program.
2. Both the implementation and experimental results are solid.
3. The paper is well-written and easy to follow.

**Weaknesses:**

1. Some claims could be more conservative.
2. Experiments are limited to a few hardware platforms.
3. It would strengthen the paper to compare with more recent related works.

**Questions:**

Thank you to the authors for submitting to ICLR 2026!

I have been following TileLang for quite some time. It is a solid piece of work that makes meaningful academic contributions and provides a robust implementation suitable for real-world model deployment. In my view, TileLang may be an even better fit for top systems conferences like ASPLOS or OSDI, where the strong implementation and in-depth design details could be more fully appreciated.

I have the following suggestions for improving the paper:

**1. Be more conservative with claims**

In the introduction, the authors use the MLA workload on H100 to highlight TileLang’s advantages over Triton, showing TileLang as over 5x faster. Other workloads reportedly achieve up to 10x speedups. While Triton can indeed be suboptimal in many cases, performance differences of several times are uncommon. The current phrasing may give readers the impression that TileLang consistently outperforms Triton by a large margin. I suggest reporting average speedups and explaining the reasons behind the 5x improvement on Hopper’s MLA workload—such as potential issues in the Triton program implementation or current compiler limitations.

**2. Add experiments on more hardware**

The evaluation primarily focuses on H100 and MI300. To demonstrate the generality of TileLang, it would be valuable to include benchmarks on other widely used architectures like Ampere and Blackwell.
For example, I previously - two or three months ago - benchmarked MHA implementations of TileLang and Tilus [1] on an RTX 4090 (q=k=v=4096, qkv heads=32, dim=128, using the example code from both GitHub projects). TileLang achieved 128 TFLOPs, while Tilus reached 148 TFLOPs. My motivation was to test whether TileLang’s pipelined loop effectively handles the prefetching optimization used in FlashAttention, which differs from that in matrix multiplication.

**3. Compare with more recent related works**

Triton’s programming model is elegant and easy to use, but its abstraction level is higher than that of the underlying hardware. As hardware evolves, achieving peak performance increasingly requires complex optimizations (e.g., software pipelining, warp specialization), which are difficult for the Triton compiler to apply automatically. Consequently, several tile-based DSLs have been proposed that sit between CUDA C and Triton in abstraction level.
To my knowledge, these include Triton Gluon [2], Tilus [1], and TileLang (the subject of this paper). A qualitative comparison among these DSLs would strengthen the paper. If possible, adding quantitative results -- even preliminary ones -- would make the comparison more compelling.

**Minor Suggestion**

Consider adding a short section acknowledging TileLang’s implementation foundation in TVM, to give appropriate credit.

**Summary**

Overall, TileLang is a novel and well-executed work. I strongly support acceptance and would champion this paper. Happy to increase the score if above concerns are addressed.

**References**

- [1] Tilus: A Tile-Level GPGPU Programming Language for Low-Precision Computation (https://arxiv.org/abs/2504.12984, https://github.com/NVIDIA/tilus)
- [2] Triton Gluon: https://github.com/triton-lang/triton/tree/main/python/tutorials/gluon

---

> ### Author Response · Authors · 2025-11-20
> **Response to Reviewer CKBd**
>
> We sincerely thank the reviewer for the very positive evaluation of TileLang. We also greatly appreciate the constructive suggestions. We have carefully addressed each concern in the revised submission, including clarifications in the introduction, expanded comparisons in the related work, additional experimental details, and improvements to technical explanations. We are very grateful if the reviewer finds the revisions satisfactory and would consider increasing the score.
>
> ## Q1. Be more conservative with claims
>
> We thank the reviewer for this valuable suggestion. Based on the reviewer's feedback, we have revised our performance claims to use range-based reporting instead of “up to” phrasing, presenting results more conservatively and accurately across all operators. For example, for FlashMLA on H100, we now report speedups of 4.06–10.59× over Triton; for Low-Bit Matmul on MI300X, we report 4.78–11.56×; for Convolution on H100, we report 1.10–1.97×; for Block Sparse Attention on H100, we report 3.42–7.87×; and for Vertical Slash Sparse Attention on H100, we report 1.16–1.97× over Triton. All other workloads have been similarly updated throughout the paper to provide readers with a more complete view of performance variations across different configurations and problem sizes.
>
> ## Q2. Add experiments on more hardware
>
> We thank the reviewer for sharing this benchmark and for suggesting broader architectural coverage. To demonstrate generality, we now include FlashAttention results on both RTX 4090 and H100, and explicitly compare against the latest Tilus implementation.
>
> On RTX 4090, TileLang achieves state-of-the-art performance among baselines and outperforms the latest Tilus by 3%–4%. The discrepancy with the reviewer’s earlier numbers is likely due to autotuning: TileLang with heuristic configurations may be suboptimal, while autotuning identifies more efficient schedules. Even without TMA on Ampere, TileLang leverages multi-buffer pipelining and layout inference to maintain strong performance.
>
> On H100, TileLang outperforms the latest Tilus by 59%–75%, owing to automatic exploitation of warp specialization, TMA, and WGMMA on Hopper.
>
> These results show that TileLang generalizes across NVIDIA architectures and consistently outperforms the latest Tilus implementation on both Ampere and Hopper.
>
> ### RTX 4090 (batch=16, heads_q=heads_kv=32, dim=128, causal=True)
>
> | name | seq_len | FA2 | Triton | Tilus | TileLang |
> | --- | ---: | ---: | ---: | ---: | ---: |
> | mha | 1024 | 137.58 | 106.61 | 133.82 | 143.40 |
> | mha | 2048 | 151.46 | 121.44 | 154.36 | 159.03 |
> | mha | 4096 | 162.70 | 129.75 | 159.26 | 162.79 |
> | mha | 8192 | 164.10 | 134.40 | 161.60 | 165.56 |
>
> Lines of Code (LoC):
>
> | FA2 | Triton | Tilus | TileLang |
> | ---: | ---: | ---: | ---: |
> | 389 | 197 | 393 | 138 |
>
> ### H100 (batch=64, heads_q=heads_kv=64, dim=128, causal=True)
>
> Tilus LOC: 83; TileLang LOC: 66
>
> | name | seq_len | Tilus (ms) | TileLang (ms) |
> | --- | ---: | ---: | ---: |
> | mha | 1024 | 6.85 | 4.29 |
> | mha | 2048 | 25.65 | 15.69 |
> | mha | 4096 | 97.14 | 55.44 |

---

> ### Author Response · Authors · 2025-11-20
> **Response to Reviewer CKBd**
>
> ## Q3. Compare with more recent related works
>
> We thank the reviewer for this helpful suggestion. As noted in our response to Q2, we already provide a quantitative comparison with Tilus. Following the reviewer’s recommendation, we have additionally included a qualitative and quantitative comparison with two other recent tile-based DSLs, Gluon and Helion.
>
> We evaluate the latest official Helion and Gluon examples that support execution on Hopper, covering both GEMM and Mamba-chunk-scan workloads. The results show that TileLang achieves 1.10–1.19× speedup over Helion and 1.53–1.83× speedup over Gluon.
>
> The differences are attributable to design limitations in existing DSLs. Helion lacks an effective tile-recommendation system, making optimization difficult and causing long tuning times (over 20 minutes). Gluon adopts a suboptimal pipeline schedule that creates substantial warp stalls, leaving kernels latency-bound. In contrast, TileLang’s tile-recommendation mechanism efficiently identifies good tile configurations, and its pipeline-inference strategy generates an effective schedule that overlaps computation and memory I/O.
>
> These additions provide a clearer qualitative and quantitative comparison among recent DSLs and further highlight TileLang’s performance and usability advantages.
>
> ### GEMM: TileLang vs. Gluon (Gluon LOC=68, TileLang LOC=17)
>
> | Config (M,N,K) | TileLang time (ms) | TileLang TFLOPS | Gluon time (ms) | Gluon TFLOPS |
> | --- | ---: | ---: | ---: | ---: |
> | 8192,1024,8192 | 0.16 | 840.09 | 0.30 | 461.85 |
> | 8192,8192,8192 | 1.40 | 784.80 | 2.22 | 496.32 |
> | 8192,28672,8192 | 5.09 | 755.42 | 7.82 | 492.03 |
> | 8192,8192,28672 | 5.09 | 756.05 | 7.77 | 495.28 |
>
> ### Mamba2 chunk-scan: TileLang vs. Helion (batch=8, heads=80, dim=64, dstate=128)
>
> Helion LOC=116, TileLang LOC=114.
>
> | seq_len | TileLang (ms) | Helion (ms) | TileLang TFLOPS | Helion TFLOPS |
> | ---: | ---: | ---: | ---: | ---: |
> | 1024 | 0.17 | 0.19 | 126.48 | 114.29 |
> | 2048 | 0.33 | 0.36 | 130.19 | 118.68 |
> | 4096 | 0.65 | 0.71 | 133.05 | 121.41 |
> | 8192 | 1.28 | 1.41 | 134.36 | 122.26 |
> | 16384 | 2.53 | 3.01 | 135.71 | 114.06 |
> | 32768 | 5.08 | 5.56 | 135.38 | 123.66 |
>
>
> ## Q4. Acknowledge TVM backend
>
> We thank the reviewer for the suggestion. We added a brief section acknowledging that TileLang is built on top of TVM. TileLang’s contributions, however, lie above the backend: a tile-level Python DSL, the fused tile-level IR (FTG), and the hardware-aware recommendation and inference pipeline. Programs lower from the TileLang DSL to FTG and then to TensorIR, where we reuse TVM’s CUDA/ROCm code generation. In summary, TileLang provides the tile abstraction and optimization pipeline, while TVM supplies the low-level backend.

---

> > ### Comment · Reviewer_CKBd · 2025-11-20
> >
> > Thanks to the authors for the detailed response! The updates regarding the claims' conservatism, the Tilus comparison on MHA (4090), and the comparison with Gluon and Helion are all satisfactory and greatly strengthen the manuscript.
> >
> > I have reviewed the revised material and have the following follow-up questions and suggestions for further clarification:
> >
> > ### 1. Automatic Pipelining of TileLang
> >
> > The strong MHA performance on the Ampere architecture suggests that the automatic software pipelining pass effectively handles both standard GEMM and more complex MHA workloads.
> >
> > * For GEMM, the standard pipelining requires preloading tiles for subsequent iterations of the outer loop (e.g., $K$-dimension).
> > * For MHA (specifically, the $Q \cdot K^T \rightarrow \text{Softmax} \rightarrow \text{Attention} \cdot V$ loop), there are often two MMAs within a single loop iteration. The necessary prefetching logic is different: we need to preload the input for the subsequent MMA *while* performing the current MMA.
> >
> > Could you please confirm that TileLang's automatic pipelining mechanism supports both of these distinct cases? If so, a brief clarification on the semantics of the `num_stages` parameter for the MHA case would be helpful, as it's not entirely clear what it means in MHA's style of pipelining.
> >
> > ### 2. Hopper Experiments on Tilus
> >
> > I noted that the Tilus team's current development [roadmap](https://github.com/NVIDIA/tilus/issues/49) (e.g., in their public GitHub issues) prioritizes B200 support, with full, optimized support for Hopper (e.g., WGMMA, TMA) and comprehensive example kernels planned for a later phase.
> >
> > Therefore, the reported H100 (Hopper) speedup over Tilus may not reflect a long-term, apples-to-apples comparison, as Tilus is not yet fully optimized for this architecture. To provide a fairer comparison and better highlight TileLang's competitive edge on currently well-supported hardware:
> >
> > * I suggest the authors focus the Tilus benchmark results on the Ampere/Ada (sm80–sm89) or Blackwell (sm100) architectures.
> > * The H100 results can be retained, but the discussion should explicitly mention Tilus's current lack of optimization for Hopper features like WGMMA and TMA, as this context is crucial for interpreting the large speedup.
> >
> > ### 3. The Performance of Gluon on GEMM
> >
> > The results showing Gluon's H100 GEMM performance at roughly half of TileLang's TFLOPS are surprising, as Gluon sits at a lower abstraction level than Triton and reuses the same backend, suggesting potential for higher peak performance.
> >
> > The authors attributed this difference to the tile-configuration recommendation. Did the authors attempt to test Gluon using the known optimal tile sizes that are typically used in Triton/TileLang kernels (e.g., a hand-picked configuration)? This would help isolate whether the bottleneck is purely the tile-recommendation system or if it is a deeper issue.
> >
> > Do the authors have any other insights into why Gluon is significantly slower in this fundamental workload?
> >
> > ### 4. Versioning of Baselines and TileLang
> >
> > For the sake of reproducibility, which is critical for systems papers, it would be better to explicitly list the version numbers (or commit hashes, if bleeding-edge) of all baselines (Tilus, Gluon, Helion, Triton) and the TileLang implementation used in evaluation.

---

> > > ### Author Response · Authors · 2025-11-21
> > > **Response to Reviewer CKBd**
> > >
> > > ### C1. Automatic Pipelining of TileLang
> > > TileLang’s automatic software pipelining mechanism supports both standard GEMM-style pipelining and the multi-MMA pattern that appears in MHA. The `num_stages` parameter provides a unified interface for pipeline construction within a loop: it specifies how many loop iterations are overlapped in the software pipeline, i.e., the depth of concurrency across iterations.
> > >
> > > For GEMM, `num_stages` controls the software pipelining depth along the reduction (`K`) dimension. It determines how many `K`-iterations are simultaneously “in flight,” following the classic pattern where the MMA compute of the current iteration overlaps with the prefetching of tiles for the next iteration (e.g., double-buffering when `num_stages = 2`).
> > >
> > > For MHA, where each iteration often contains two MMAs (e.g., `q@k` and `p@v`) and two memory load operations (e.g., load `K` and load `V`), `num_stages` controls the software pipelining depth along `K/V`’s sequence length. Setting `num_stages = 2` means that the computation of the current iteration (both MMAs) is overlapped with the loading of the `K/V` tiles needed by the next iteration, so that memory transfers for the next step are largely hidden behind the current step’s compute.
> > >
> > > TileLang analyzes buffer usage and dependencies to automatically construct the appropriate pipeline schedule for both cases.
> > >
> > > ### C2. Hopper Experiments on Tilus
> > >
> > > We thank the reviewer for the insightful suggestion. As noted in our RTX 4090 results (in response to Q2), TileLang is only slightly ahead of Tilus on Ampere/Ada (sm80–sm89), which aligns with the reviewer’s observation.
> > >
> > > Following this suggestion, we have updated Sec. 5.4 to explicitly state that Tilus is not yet fully optimized for Hopper features such as WGMMA and TMA, and we also mention that TileLang is slightly better than Tilus on Ampere/Ada.
> > >
> > > We also add the Tilus benchmark results on Ampere/Ada in Appendix L.
> > >
> > > ### C3. The Performance of Gluon on GEMM
> > > We thank the reviewer for the insightful questions. For the tiling configuration, our evaluation follows Gluon’s official testing script ([link](https://github.com/triton-lang/triton/blob/main/python/tutorials/gluon/05-wgmma.py)), which includes its built-in auto-tuning procedure. We also verified that the search space contains well-known high-performance tile sizes, such as `(128, 256, 128)`, and our measured results are consistent with those produced by the official script.
> > >
> > > Regarding why Gluon remains significantly slower, our view is that it lacks a compiler-visible tile abstraction and does not provide interfaces for automated pipeline scheduling. This forces users to manually coordinate memory access, compute overlap, and synchronization, which is error-prone and difficult to optimize consistently. In contrast, TileLang performs automatic pipeline inference and scheduling based on a compiler-visible tile abstraction, reducing developer burden and yielding more reliable performance. This gap is also reflected in code size: Gluon’s GEMM requires 68 lines of code, whereas TileLang’s implementation uses only 17.
> > >
> > > ### C4. Versioning of Baselines and TileLang
> > > We thank the reviewer for the helpful suggestion. Following this advice, we have added the exact version numbers (or commit hashes for bleeding-edge builds) for all baselines (Tilus, Gluon, Helion, Triton) as well as the TileLang implementation used in our evaluation. These details are now included in the new Appendix M of the revised paper.

---

> > > > ### Comment · Reviewer_CKBd · 2025-11-21
> > > >
> > > > Thanks the authors response and revisions!
> > > >
> > > > ### 1. Automatic Pipelining Granularity
> > > > I have one followup question regarding the automatic pipelining. The current pipelining algorithm seems only work at the loop iteration granularity. The prefetching of MHA used in some manual written MHA kernel like Tri Dao's [implementation](https://github.com/Dao-AILab/flash-attention/blob/main/csrc/flash_attn/src/flash_fwd_kernel.h) is slightly different. The workflow looks like:
> > > >
> > > > ```
> > > >   - ...
> > > >   - MMA0[i], Load V[i]
> > > >   - MMA1[i], Load K[i+1]
> > > >   - MMA0[i+1], Load V[i+1]
> > > >   - MMA1[i+1], Load K[i+2]
> > > >   - ...
> > > > ```
> > > >
> > > > The TileLang's generated pipeline looks like:
> > > >
> > > > ```
> > > >   - ...
> > > >   - MMA0[i] -> MMA1[i], Load V[i+1], K[i+1]
> > > >   - MMA0[i+1] -> MMA1[i+1], Load V[i+2], K[i+2]
> > > >   - ...
> > > > ```
> > > > The advantage of TileLang's pipeline is that the two MMA's computation provides longer time to hide the prefetching. The disadvantage is that it requires double-size shared memory for K and V.
> > > >
> > > > My question are: 1) Do you think it's valuable to support the first-style of automatic prefetching? As it's widely used. 2) Is it hard to support?
> > > >
> > > > ### 2. Design Tradeoff of TileLang
> > > >
> > > > In fact, I have a more general question and would like to learn your thoughts: for tile-level GPU programming language, what should be done by the compiler automatically and what should leave to the kernel developer to write? There are a bunch of optimizations/choices:
> > > >
> > > > * Tile size
> > > > * Automatic software pipelining
> > > > * Automatic warp specialization
> > > > * Layout of register tile, shared memory, (maybe) tensor memory (in blackwell)
> > > > * ...
> > > >
> > > > What's the advantage/disadvantage of different design choices? What's TileLang's design philosophy regards this tradeoff?

---

> ### Author Response · Authors · 2025-11-23
> **Response to Reviewer CKBd**
>
> ## 1. Automatic Pipelining Granularity
>
> We thank the reviewer for their insightful observations.
>
> First, we fully support the pipeline schedule provided by the reviewer. The workflow in Tri Dao’s implementation is equivalent to the following form:
>
> ```
> Load K[i]; MMA0[i]
> Load V[i]; MMA1[i]
> Load K[i+1]; MMA0[i+1]
> Load V[i+1]; MMA1[i+1]
> ...
> ```
>
> The only difference is that in our formulation `Load K[i+1]` is placed at the beginning of the next iteration, but the execution order is semantically equivalent. Consequently, TileLang can naturally support the same pattern, for example:
>
> ```python
> for i in T.pipelined(loop_range, num_stage=1):
>     Load K[i]
>     MMA0[i]
>     Load V[i]
>     MMA1[i]
> ```
>
> In this case, there is no need to double-buffer K/V.
>
> Second, beyond exposing a simple unified parameter such as `num_stage`, the `Pipelined` interface in TileLang also provides finer-grained scheduling controls (e.g., `order`, `stage`) to specify the position of each statement within the loop body. In addition, TileLang allows users to manually control the placement and specialization of individual statements (e.g., which warps execute them), as well as the acquisition and release of barriers. This makes pipeline control more flexible and expressive, not restricted to loop-iteration–level granularity.
>
> For example, in our FA3 implementation, TileLang constructs a deeper pipeline across the MMA, softmax, and load operations by using a schedule of the following form, where the `order` and `stage` arrays specify, respectively, the execution order of the five statements within the loop body and the loop iteration to which each statement is assigned:
>
> ```python
> for i in T.pipelined(loop_range, order=[3, 4, 0, 2, 1], stage=[0, 0, 1, 1, 2]):
>     Load K[i]
>     Load V[i]
>     MMA0[i]
>     Softmax[i]
>     MMA1[i]
> ```
>
> which is lowered to an execution order conceptually equivalent to:
>
> ```python
> for i in range(loop_range):
>     MMA0[i + 1]
>     MMA1[i]
>     Softmax[i + 1]
>     Load K[i + 2]
>     Load V[i + 2]
> ```
>
> thereby achieving finer-grained overlapping between loading, computation, and softmax.
>
> Similarly, in our sparse MLA implementation we leverage the ability to manually control the placement and specialization of individual statements, for instance with a pattern such as:
>
> ```python
> for i in range(loop_range):
>     if tx < 256:
>         T.wait(data_ready)
>         MMA0(...)
>         MMA1(...)
>         T.arrive(data_release)
>     else:
>         T.wait(data_release)
>         Load K(...)
>         T.arrive(data_ready)
> ```
>
> Here, different warps (distinguished by `tx`) are assigned distinct roles (compute vs. load), and wait/arrive barriers are used to coordinate them, enabling warp-specialized, fine-grained pipelining beyond simple loop-iteration scheduling.
>
>
> ## 2. Design Tradeoff of TileLang
>
> We appreciate this question, as it gets at a core design tradeoff for tile-level DSLs. Existing systems indeed make different choices along this spectrum: Triton largely pushes tile-level optimizations to the compiler (e.g., tile sizes, layouts, software pipelining), while Gluon expects the kernel developer to manually orchestrate many of these decisions.
>
> TileLang is explicitly designed around a flexible tradeoff between these two extremes. It exposes tile-level optimization dimensions as explicit, tunable parameters (e.g., tile size, software pipelining, warp specialization, and memory layout), so that expert developers can override or constrain them when needed. However, all of these annotations are optional. When user annotations are absent, TileLang’s compiler uses hardware-aware inference and recommendation to choose reasonable tile sizes, layouts, and pipelining strategies in a best-effort manner, guided by hardware constraints and best practices.
>
> Our design philosophy is therefore:
>
> - The kernel developer describes the algorithmic structure and, when needed, provides high-level hints or constraints through tile-level annotations.
> - The compiler handles the mechanical, hardware-specific decisions (e.g., precise tiling factors, buffer scheduling, warp specialization, layout selection) whenever the developer does not want to micromanage them.
>
> This hybrid model aims to combine the productivity and portability of a Triton-like “compiler-does-most” approach with the fine-grained control and debuggability of a Gluon-like “developer-in-the-loop” approach. In practice, developers can start from TileLang’s recommended configurations and rely on automatic inference for most kernels; for performance-critical kernels on a specific architecture, they can selectively fix certain tile parameters and let the compiler complete the remaining decisions. (Also related to our response to W3 of reviewer CYSa.)

---

> > ### Comment · Reviewer_CKBd · 2025-11-25
> >
> > Thanks the author's detailed response!
> >
> > All my concerns are addressed, and happy to discuss with you.
> >
> > Will raise my score from 8 to 10, good work!

---

### Official Review · Reviewer_CYSa · 2025-11-01

**Soundness:** 4
**Presentation:** 3
**Contribution:** 2
**Rating:** 4
**Confidence:** 5

**Summary:**

This work proposes TileLang, a framework to help simplify GPU kernel programming. The framework automates decisions around memory movement and scheduling, while exposing tile primitives and Pythonic operators. The work evaluates on NVIDIA and AMD GPUs and across a breadth of workloads.

**Strengths:**

The results are strong and the kernel examples / level of simplicity is compelling. This seems like a useful system for the AI community. The baselines for evaluation are well-chosen and the chosen level of abstraction to expose to the developer (between programming primitives, compiler automations) seems effective.

**Weaknesses:**

It is not very precise how the automation works within TileLang / what the underlying algorithms are. The automation is very nice, but it's not clear if it breaks down in any settings since that's not fully analyzed/explained to the reader. The paper makes clear *what* is done, but not *how* it's accomplished.
- It is not clear how TileLang handles the memory access patterns for NVIDIA vs. AMD.
- It’s is not clear where the line is that divides the developer’s tasks from what TileLang automates.
- It’s clear that the code is compact from the plots, but it’s not explained what deltas from existing frameworks contribute to the code size delta.
- It is not clear what contributes to the speed ups over the baselines

The paper does not clearly compare to prior work, which makes it difficult to assess novelty. For instance, ThunderKittens (over a year old) is not mentioned in the related works or intro despite also using automatically optimized tile primitives and Pythonic functions. Mojo is another compiled language that works on both AMD and NVIDIA with tiles. In the intro Contribution #1 is tile abstractions, which has been shown in extensive prior work and it’s not clear how TileLang differs. Contribution #2 is tile configuration. Again prior frameworks provide default swizzle patterns and register layouts for matrix instruction alignment; how does the proposed method differ?

The writing could be clearer in a few places:
- What is the backend of TileLang? The intro doesn’t explain what language / frameworks it’s built on. For instance, the first time it’s mentioned that TileLang is a compiler/uses compiler passes is on L122.
- L122: Why are compiler analyses needed versus making the load/store functions use coalesced load patterns and loop tiling to begin with? What additional value is the compiler providing?

Results:
- Does the MI300X kernel match the performance of FlashAttention-3 on the H100? It would be useful to know the absolute TFLOP numbers everywhere, since PyTorch is a constantly changing baseline. For instance, some analysis has shown that PyTorch SDPA achieves 20-100 TFLOPs in some AMD Docker containers, so it is difficult to understand the results.
- Is there an analysis that shows that the “Pipelined” abstraction is sufficient for all warp specialization a user might want to do? I notice that synchronization is abstracted away from the user. Is there ever a case where the user would want to manage synchronization points more carefully and how does TileLang support this if so?
- Are the appendix kernels the same on NVIDIA and AMD?

Overall this is a very solid and useful system, but the research aspects -- how the philosophy and contributions are novel, and how the algorithms work -- have not been sufficiently explained.

**Questions:**

See questions above.

---

> ### Author Response · Authors · 2025-11-20
> **Response to reviewer CYSa**
>
> We thank the Reviewer for the helpful comments. Based on the reviewer’s suggestions, we have revised the paper to better explain the novelty of TileLang and to clarify how the algorithms work. We hope these changes address the reviewer’s concerns and would appreciate reconsideration of the score.
>
> ## W1. It is not very precise how the automation works within TileLang / what the underlying algorithms are
>
> We thank the reviewer for raising this point. To make the automation mechanisms more precise, we have added formal pseudocode and detailed algorithmic descriptions in Appendix E and Appendix F in the revised paper. These updates include the hardware-aware Layout Inference procedures (**Algorithms 1–3**) and the Pipeline Inference procedures (**Algorithms 4–5**). These algorithms formally explain the inference workflows illustrated in Fig. 5 and Fig. 6, making the underlying automation in TileLang explicit and precise. Also, we clarify that instruction selection is delegated to vendor tile libraries (NVIDIA CuTe/CUTLASS and AMD CK) through `T.call_extern`, which selects the appropriate instructions based on shape and dtype.
>
> We direct the reviewer to the updated Appendix for the detailed pseudocode and precise explanations.
>
> ## W2. It is not clear how TileLang handles the memory access patterns for NVIDIA vs. AMD.
>
> We thank the reviewer for raising this question. TileLang handles NVIDIA and AMD memory access through a unified approach: both platforms share broadly similar memory hierarchies (global → L2 → shared/LDS → registers/VGPRs) and SIMT execution models, so the same TileLang algorithm runs unmodified on both.
>
> To reach peak performance, TileLang combines automatic layout inference with backend-aware scheduling:
>
> - Layout inference. The compiler automatically accounts for key differences such as 32-thread warps vs. 64-lane wavefronts, Tensor Core `mma.sync` vs. `MFMA` instructions, and distinct swizzle/banking rules. It runs global inference over the FTG to ensure legal, coalesced, and bank-conflict-free layouts end-to-end (global, shared/LDS, registers). Users may override layouts when needed.
> - Scheduling. As shown in Figure 11 (Appendix D), TileLang selects different tile sizes, memory placements, pipelining strategies, and warp-partitioning policies for the H100 and MI300X, reflecting their differences in memory and register capacities as well as their distinct compute units.
>
> In short, TileLang decouples the algorithm from backend-specific layout and scheduling. The same kernel runs on both NVIDIA and AMD, while automated layout inference and small backend-specific scheduling refinements bring performance close to hand-tuned kernels. We have clarified these mechanisms and added representative configurations in the revision.
>
> ## W3. where the line is that divides the developer’s tasks from what TileLang automates.
>
> The reviewer raises an important question. In TileLang, the boundary between the developer’s tasks and what the system automates is defined by the fused tile graph (FTG). An FTG consists of nodes, edges, and node annotations. The developer is responsible for specifying the FTG topology—that is, how tile operators are composed to express the computation logic of the kernel. Tile-level annotations (e.g., tile sizes, memory placement, warp partitioning, tensorization) are optional.
>
> A partially annotated FTG then becomes the input to TileLang’s automated optimization pipeline. From this input, TileLang performs tile recommendation and tile inference, propagating shape, layout, and memory constraints across the graph to fill in the remaining annotation fields (as illustrated in Figure 3). This design clearly marks the division of responsibility:
>
> - (a) Developers define the computation logic (the FTG structure).
> - (b) TileLang completes and optimizes the annotation space automatically.
>
> The system is intentionally flexible: developers may provide high-quality starting points using TileLang’s recommendations or rely entirely on automated inference. When no annotations are given, inference proceeds in a best-effort manner guided by hardware constraints and best practices. When expert-provided starting points are available, TileLang incorporates them to more effectively refine and complete the FTG for near-peak performance.

---

> > ### Author Response · Authors · 2025-11-20
> > **Response to reviewer CYSa**
> >
> > ## W4. What deltas from existing frameworks contribute to the code size delta
> >
> > The reviewer raises an important question. TileLang’s code-size reduction primarily comes from its fundamentally different tile abstraction (see Response to W6). In Triton, Mojo, and ThunderKittens, a “tile” is just a manually managed shared-memory buffer, so developers must write element-level pointer arithmetic, cooperative loads, and inner loops by hand. The compiler only sees low-level pointer operations and cannot reason about tile structure.
> >
> > TileLang instead represents tiles as first-class IR constructs with explicit semantics for indexing, data movement, and pipelining. High-level primitives such as `copy`, `gemm`, and `pipelined` allow the compiler to automatically choose block shapes, swizzles, and pipeline stages through our two-stage recommendation and inference process.
> >
> > This abstraction directly yields the code-size delta. Since TileLang operates on tile objects rather than per-element pointers (e.g., `A[by * block_M, k * block_K]`) rather than per-element offsets), the compiler—not the programmer—handles address calculation, cooperative loads, and tile scheduling, removing most of the boilerplate required in existing frameworks.
> >
> > Concretely, two deltas dominate:
> >
> > - Tile-level indexing replaces manual pointer arithmetic.
> > - Automatic context and thread mapping remove launch and thread-index boilerplate.
> >
> > These deltas explain the compact kernels. To make these simplifications concrete, we provide a side-by-side GEMM implementation in Triton and TileLang and annotate each location where TileLang removes boilerplate relative to Triton (see **Appendix I**). We hope this helps clarify the design rationale behind the resulting brevity.
> >
> > ## W5. What contributes to the speedups over the baselines
> >
> > We thank the reviewer for the helpful suggestion. In response, we added a breakdown of the Triton baseline in the Ablation Studies section, using FlashMLA as a representative example to clarify which components of TileLang’s automation contribute to the observed speedups. On H100, tiling (+Tile) provides a modest improvement of ≈1.31×, while warp partitioning (+Partition) is the dominant contributor with ≈4.34×. On MI300X, allocation placement (+Alloc) is the dominant contributor with 6.56 ×, and tiling (+Tile) contributes 1.75 ×. This breakdown makes the sources of acceleration explicit and directly addresses the reviewer’s concern.

---

> ### Author Response · Authors · 2025-11-20
> **Response to reviewer CYSa**
>
> ## W6. Novelty of tile abstractions and tile configuration
>
> We appreciate the reviewer’s comment and clarify the relationship to prior work. TileLang’s Contribution 1 is not a reuse of existing tile primitives. In prior frameworks such as Mojo or ThunderKittens, a “tile” is essentially a manually managed shared-memory buffer: developers compute pointer offsets, orchestrate cooperative loads, and write inner loops by hand. Because the compiler only sees pointer arithmetic and unstructured loops, the tile’s structure is not represented at the IR level, which limits the system to fixed heuristics (e.g., default swizzles or fragment layouts) and prevents systematic search or transformation.
>
> TileLang introduces a fundamentally different abstraction by elevating tiles to first-class IR constructs with explicit semantics for indexing, data movement, reuse, and pipelining. These IR tiles are manipulated through high-level primitives such as `copy`, `gemm`, and `pipelined`, making tile behavior compiler-visible and enabling transformations that prior systems cannot express. This richer and more structured representation is the basis for Contribution 2: our two-stage tile-configuration framework. Because tile structure is explicit in the IR, the compiler can reason about and automatically optimize block shapes, swizzles, pipeline stages, and register layouts using the unified FTG representation, rather than relying on fixed, hand-designed defaults.
>
> In short, the feasibility and effectiveness of Contribution 2 directly stem from the novelty of Contribution 1. TileLang differs from prior systems not only in the surface syntax, but in the underlying compiler-visible tile abstraction that makes automated configuration possible.
>
> ThunderKittens (TK) is discussed in the related works, and we include direct empirical comparisons with TK in Sec. 5.2. In addition, we added a new Sec. 5.4 that provides a quantitative comparison with more recent DSLs such as Tilus, Gluon, and Helion. These results show that TileLang effectively bridges programmability and performance across modern GPUs. The details and evaluation are further discussed in our response to Reviewer CKBd–Q2/Q3.
>
> ### References
>
> [1] Ding Y, Hou B, Zhang X, et al. Tilus: A Virtual Machine for Arbitrary Low-Precision GPGPU Computation in LLM Serving. arXiv preprint arXiv:2504.12984, 2025.
>
> [2] OpenAI Triton Team, “Gluon,” https://github.com/triton-lang/triton/tree/main/python/examples/gluon, 2025.
>
> [3] PyTorch Team at Meta, Helion: A High-Level DSL for Performant and Portable ML Kernels, https://github.com/pytorch/helion, 2025.
>
> [4] Godoy W F, Melnichenko T, Valero-Lara P, et al. Mojo: MLIR-Based Performance-Portable HPC Science Kernels on GPUs for the Python Ecosystem[J]. arXiv preprint arXiv:2509.21039, 2025.
>
> ## W7. What is the backend of TileLang?
>
> We appreciate the reviewer’s question. TileLang is implemented on top of the TVM backend, but our main contributions sit above TVM. TileLang introduces a new tile-level Python DSL, a fused tile-level dataflow IR (FTG), and a hardware-aware recommendation and inference pipeline. These components define TileLang’s programming and optimization model and are not provided by TVM.
>
> In practice, TileLang programs are written using compiler-visible tile primitives (such as `T.alloc`, `T.copy`, and `T.gemm`). The DSL lowers into FTG, which carries tile shapes, swizzles, pipeline, and partition decisions. Once FTG is legalized, it is lowered into TensorIR, where we reuse TVM’s mature CUDA/ROCm code generation and runtime.
>
> Thus, TileLang provides the tile abstraction, FTG IR, and optimization passes, while TVM supplies the low-level backend. We have updated the paper in Figure 7 to introduce this workflow (TileLang DSL → FTG → TensorIR).

---

> ### Author Response · Authors · 2025-11-20
> **Response to reviewer CYSa**
>
> ## W8. L122: Why are compiler analyses needed? What additional value is the compiler providing?
>
> We appreciate the reviewer’s question. At L122, the compiler analyses provide additional value that user-written load/store templates cannot reliably achieve. TileLang does allow users to specify coalesced load/store patterns or manual loop tiling, but this quickly becomes error-prone: hand-crafted patterns can violate dependences, create misaligned or bank-conflicting accesses, mishandle irregular tile boundaries, or exceed register and shared-memory budgets. These issues grow significantly in fused or irregular kernels.
>
> With the tile abstraction and explicit thread context, the compiler has full visibility into tile shapes and thread/block geometry. This allows it to synthesize correct and efficient coalesced access patterns, select legal tiling strategies, insert asynchronous copies and synchronizations, and generate the necessary boundary predicates. The analyses also coordinate global legality checks across dependences, memory layouts, vectorization widths, resource usage, and occupancy. As a result, the final kernel remains correct, efficient, and portable across hardware variants.
>
> Therefore, compiler analyses are essential because they provide:
> - Correctness and legality across dependences and boundaries
> - Portability across hardware variants
> - Resource-aware optimization (registers, shared memory, occupancy)
> - Coordinated scheduling of copies, synchronization, and vectorization
>
> ## W9. Does the MI300X kernel match the performance of FlashAttention-3 on the H100? It would be useful to know the absolute TFLOP numbers
>
> We appreciate the reviewer’s suggestion. For example, on MI300X, the state-of-the-art FlashAttention kernel in Composable Kernel (CK) achieves an average of 446 TFLOPs. On H100, the official FlashAttention-3 implementation achieves an average of 565 TFLOPs, indicating that MI300X delivers 78.9% of H100 performance on this workload.
>
> TileLang on MI300X reaches an average of 423 TFLOPs, which corresponds to 94.8% of CK performance. All TFLOP values are computed using the standard definition based on the number of 16-byte floating multiply-add operations in the QK and PV phases of the attention computation.

---

> ### Author Response · Authors · 2025-11-20
> **Response to reviewer CYSa**
>
> ## W10. Is there an analysis that shows that the “Pipelined” abstraction is sufficient for all warp specialization a user might want to do? Is there ever a case where the user would want to manage synchronization points more carefully and how does TileLang support this if so?
>
> The reviewer raises a very good point. Our `Pipelined` abstraction is designed to capture the common producer–consumer form of warp specialization, where one warp group issues asynchronous copies and the others compute. In this setting, the compiler can automatically place waits and barriers, insert async copies, and enforce legality and resource constraints. In our H100 evaluation, this abstraction was sufficient for FlashAttention-like and matmul-like kernels, which represent the majority of practical warp-specialized workloads.
>
> At the same time, TileLang does not restrict users to this high-level form. For cases where developers need finer control over synchronization, TileLang exposes explicit primitives, including `T.alloc_barrier`, `T.barrier_arrive`, and `T.barrier_wait`, as well as user-specified pipeline depth and warp-partitioning annotations (e.g., `T.ws(...)`). These features allow users to write fully manual warp-specialized kernels when needed.
>
> Below is an example demonstrating how TileLang supports explicit barrier-based synchronization alongside the `Pipelined` structure:
>
> ```python
> for ko in T.Pipelined(T.ceildiv(K, block_K), num_stages=2):
>     with T.ws(1):  # producer warp group
>         T.barrier_wait(compute_is_done, (ko + 1) % 2)
>         T.copy(A[by * block_M, ko * block_K], A_shared)
>         T.copy(B[ko * block_K, bx * block_N], B_shared)
>         T.barrier_arrive(data_is_ready)
>     with T.ws(0):  # consumer warp group
>         T.barrier_wait(data_is_ready, ko % 2)
>         T.gemm(A_shared, B_shared, C_local)
>         T.barrier_arrive(compute_is_done)
> ```
>
> This example illustrates how TileLang supports both the high-level `Pipelined` warp-specialization pattern and explicit, user-controlled synchronization when finer control is required. We will include similar examples in the open-source release.
>
> ## W11. Are the appendix kernels the same on NVIDIA and AMD?
>
> The reviewer raises a good point. The appendix kernels use the same TileLang source on both NVIDIA and AMD, and the identical code compiles and runs on both backends. The standard TileLang kernel is therefore exactly the same across platforms.
>
> For peak performance, TileLang automatically accounts for vendor differences and allows small backend-specific refinements without changing the algorithm. These include choosing appropriate fragment layouts and swizzles for Tensor Cores and MFMA units, and making slight adjustments in tile sizes or memory placement.
>
> We also added Appendix K, which shows a side-by-side comparison of the high-performance AMD and NVIDIA FlashAttention kernels, illustrating that the high-level TileLang code remains almost identical while still achieving efficient performance.

---

### Official Review · Reviewer_Bjv9 · 2025-11-01

**Soundness:** 4
**Presentation:** 3
**Contribution:** 4
**Rating:** 8
**Confidence:** 5

**Summary:**

This paper introduces TILELANG, a novel programming system that bridges the programmability-performance gap in AI accelerator programming. It treats "tiles" (hyper-rectangular tensor slices) as first-class entities in a Pythonic DSL, enabling explicit control over memory, data movement, and scheduling. Core innovations include:

Tile Recommendation: Uses a roofline model to suggest hardware-aware defaults (tile sizes, memory placement, warp partitioning).
Tile Inference: Autocompletes low-level details (memory layouts, schedules) via constraint propagation from partial specifications or recommendations.
On NVIDIA H100 and AMD MI300X GPUs, TILELANG matches or exceeds hand-tuned libraries (FlashAttention, CUTLASS) and outperforms Triton, with code size closer to manual implementations but simpler.

**Strengths:**

Solves a Critical Problem: Precisely targets the core trade-off between programmability and peak performance in AI kernel development, a highly relevant issue.

Innovative Human-in-the-Loop Framework: The novel "Tile Recommendation + Tile Inference" model is highly effective, lowering the barrier for entry while retaining expert control and drastically reducing manual tuning effort.

Elegant Programming Model: The tile-centric DSL is clean and expressive, allowing developers to control low-level hardware behavior (memory, parallelism) in a high-level language with concise code.

Highly Convincing Evaluation: Achieves state-of-the-art (SOTA) or near-SOTA performance on both NVIDIA and AMD GPUs with significantly less code than hand-tuned libraries, demonstrating its effectiveness and cross-platform capability.

**Weaknesses:**

1. The roofline-based analytical model ignores complex microarchitectural details, which could lead to suboptimal suggestions in certain scenarios.

2. There are several works  about kernel generators that are not cited properly.

**Questions:**

1. What performance trade-offs were made to create a unified API for both NVIDIA and AMD? Does this abstraction hide hardware-specific features that might be key for hitting peak performance on a single platform?

2. How would TILELANG's tile-based abstractions extend to highly irregular computations, such as sparse operations or Graph Neural Networks, where data access patterns are not uniform?

3. The appendix reports impressive tuning times of around 10-15 seconds. However, this is not directly compared to the tuning time of other auto-tuning frameworks (e.g., TVM's Ansor/AutoTVM, Triton's autotuner) for the same tasks. A direct comparison would better highlight the efficiency advantage of the recommendation/inference approach over empirical search.

---

> ### Author Response · Authors · 2025-11-20
> **Response to Reviewer Bjv9**
>
> Thanks for your careful review and feedback! All the comments are helpful.
>
> ## W1. The roofline-based analytical model
>
> The reviewer raises a good point. We acknowledge that analytical models involve design tradeoffs. Consistent with the tradeoff made by the tile abstraction itself (see Response to Q1), our cost model adopts the same principle: it reasons at the tile granularity, which is the level where performance-critical microarchitectural behaviors actually emerge. Rather than relying on a simple roofline approximation, the model is tile-driven and therefore able to capture memory-hierarchy interactions, heterogeneous compute-unit utilization, cache-miss tendencies, and scheduling overheads.
>
> In Equation (1), we explicitly model the memory hierarchy (HBM/L2/L1 indexed by i) and distinguish tensor cores, vector cores, and SFUs (indexed by j), each evaluated using tile-level execution metrics. Tile load order and data reuse are estimated using reuse-distance analysis, and the intrinsic term captures non-divisible waves and pipeline prologue/epilogue overheads. This design keeps the model lightweight while achieving high predictive accuracy, pruning 95 percent of configurations while preserving 98.47 percent of peak performance (Appendix G).
>
> The tradeoff, consistent with the tile abstraction, appears below the tile level. Extremely fine-grained behaviors beneath the tile granularity are not modeled, such as undocumented scheduling rules or hidden register and shared-memory allocation effects introduced during CUDA to PTX/SASS lowering. These inaccessible micro-details explain the remaining gap between TileLang-generated kernels and highly hand-tuned PTX implementations such as FlashMLA.
>
> ## W2. There are several works about kernel generators that are not cited properly.
>
> We thank the reviewer for noting that several kernel-generation systems were not properly cited. We have revised the Related Work to explicitly include and discuss Ansor [zheng2020ansor], PET [wang2021pet], TensorIR [feng2023tensorir], and Mirage [wu2024mirage], and we clarified TileLang’s relation to them.
>
> These systems rely on schedule search (Ansor), partially equivalent transformations with repair (PET), generalized loop-nest IRs (TensorIR), or hierarchical GPU abstractions (Mirage). TileLang differs by adopting a tile-first design that automates layout and configuration while still exposing explicit control over fusion, memory hierarchy, parallelism, and warp partitioning. This enables precise expression and optimization of highly performance-sensitive kernels such as FlashAttention and MLA.
>
> We appreciate the reviewer’s suggestion, and the revised Related Work reflects these additions. The additions are shown in the Appendix A
>
> [1] L. Zheng et al. Ansor: Generating High-Performance Tensor Programs for Deep Learning. OSDI ’20.
>
> [2] H. Wang et al. PET: Optimizing Tensor Programs with Partially Equivalent Transformations and Automated Corrections. OSDI ’21.
>
> [3] S. Feng et al. TensorIR: An Abstraction for Automatic Tensorized Program Optimization. ASPLOS ’23.
>
> [4] M. Wu et al. Mirage: A Multi-Level Superoptimizer for Tensor Programs. OSDI ’25.
>
> [5] J. Zhao et al. AKG: Automatic Kernel Generation for Neural Processing Units Using Polyhedral Transformations. PLDI ’21.

---

> ### Author Response · Authors · 2025-11-20
> **Response to Reviewer Bjv9**
>
> ## Q1. Performance trade-offs in abstractions
> The reviewer raises another good point. Our unified API does introduce trade-offs, but these are made at the tile granularity, which we found to be the right level for supporting both NVIDIA and AMD without sacrificing performance-critical expressiveness. The unified API does not hide hardware-specific features. Instead, it explicitly models the memory hierarchy (HBM, L2, L1) and the major compute and data-movement units common to modern GPUs across vendors. These are the key factors that must be controlled to reach near-peak performance.
>
> TileLang exposes these tile-level optimization dimensions as tunable parameters, including tile size, memory placement, warp partitioning, memory layout, software pipelining, and tensorization. These parameters act as hardware-aware knobs that allow the compiler to specialize for each architecture. Under the unified interface, TileLang performs platform-specific recommendation and inference. As shown in Figure 11 (Appendix D), TileLang selects different tile sizes, memory placements, pipelining strategies, and warp-partitioning policies for the H100 and MI300X, reflecting their differences in memory and register capacities as well as their distinct compute units.
>
> The actual trade-off appears below the tile level. TileLang does not attempt to model extremely fine-grained behaviors beneath the tile granularity, such as undocumented scheduling effects or hidden register and shared-memory allocation decisions introduced during CUDA to `PTX` or `SASS` lowering. These details are difficult to expose in a stable and portable API, and they explain the remaining performance gap between TileLang-generated kernels and fully hand-crafted PTX-level implementations such as FlashMLA.
>
> ## Q2. Highly irregular computations
> The reviewer raises a good question. TileLang’s tile abstraction extends naturally to irregular and sparse workloads through its index-in-tile programming mechanism, which allows arbitrary per-element indexing inside a tile. This provides native support for non-uniform and data-dependent access patterns without modifying the underlying abstraction.
>
> For example, the sparse DSA pattern used in DeepSeek-V3.2 can be expressed by simply replacing a dense MLA load such as `KV_shared[i, d] = KV[b, s, g, d]` with an indirect-indexed variant `KV_shared[i, d] = KV[b, Indices[b, s, g, i*BI + b], g, d]`. This preserves the simplicity of tile-level programming while enabling flexible expression of sparse and irregular memory-access behaviors. We will release the DSA implementation as part of our open-source code.
>
> More broadly, the same index-in-tile mechanism applies directly to graph-structured operators. Since GNN primitives such as neighbor aggregation, scatter/gather, and message passing are fundamentally index-driven, they can be implemented in TileLang using the same tile-level model and indirect-access patterns.

---

> ### Author Response · Authors · 2025-11-20
> **Response to Reviewer Bjv9**
>
> ## Q3. Compared to auto-tuning frameworks  (e.g., TVM's Ansor/AutoTVM, Triton's autotuner)
> We thank the reviewer for the suggestion. We have added a direct tuning-time comparison with Ansor/AutoTVM and Triton for the GEMM kernel on H100. The results show that TileLang tunes markedly faster than both frameworks—especially vs. TVM Ansor, which requires much longer empirical search. This improvement comes from TileLang’s first-class tile IR, which defines a far more structured optimization space, and from our cost-model–guided inference, which avoids large brute-force searches.
>
>
> #### GEMM@H100
>
> | M,N,K                | TileLang tuning time (s) 20 cfgs | Triton time (s)20 cfgs | TVM (s)100 trials |
> |----------------------|:-----------------------------------:|:--------------------------:|:---------------------:|
> | 8192,1024,8192       | 11.81                              | 18.43                     | 518.52                |
> | 8192,8192,8192       | 11.78                              | 18.24                     | 455.51                |
> | 8192,28672,8192      | 14.59                              | 20.15                     | 3007.00               |
> | 8192,8192,28672      | 14.31                              | 20.07                     | 4142.05               |
>
> #### Conv2@H100
>
> | N, C, H, W, F, K, S               | TileLang tuning time (s)20 cfgs | Triton time (s)20 cfgs |
> |------------------------------------|:-----------------------------------:|:--------------------------:|
> | 128, 2048, 7, 7, 512, 1, 1         | 11.56                              | 17.56                     |
> | 128, 512, 7, 7, 2048, 1, 1         | 17.49                              | 18.76                     |
> | 128, 512, 14, 14, 512, 3, 2        | 17.65                              | 37.59                     |
> | 128, 512, 7, 7, 512, 3, 1          | 17.41                              | 38.57                     |
> | 128, 256, 14, 14, 256, 3, 1        | 19.18                              | 37.61                     |
> | 128, 1024, 14, 14, 256, 1, 1       | 18.87                              | 18.82                     |
> | 128, 512, 28, 28, 128, 1, 1        | 17.50                              | 19.13                     |
> | 128, 64, 56, 56, 64, 1, 1          | 17.20                              | 18.97                     |

---

> > ### Comment · Reviewer_Bjv9 · 2025-11-23
> >
> > Thanks for the authors' response. I continue to recommend acceptance of this paper.

---

### Author Response · Authors · 2025-12-01
**Summary comment by Authors**

Thank you for serving as our AC and for the constructive feedback and discussion from the reviewers. In our responses, we addressed all comments in detail and updated the manuscript accordingly, with changes highlighted in blue.

Our paper proposes **TileLang, a tile‑centric DSL and compiler** for developing high‑performance AI kernels across accelerators.
- Reviewers broadly agree that the work addresses **an important and timely problem**: making it easier to write peak‑performance AI kernels on increasingly complex GPUs.
- Reviewers BJV9 and 7JBZ praise the tile‑centric DSL and human‑in‑the‑loop workflow (tile recommendation and inference) as **an innovative, principled way** to balance automation with expert control.
- Reviewer CYSa views TileLang as **a useful system** for the AI community that provides an effective abstraction for developers.
- Reviewer 7JBZ considers TileLang as an excellent contribution from **both research and engineering perspectives**.
- All reviewers find the **implementation and experiments solid**,  with TileLang achieving SOTA or near-SOTA performance using significantly less code.

In the rebuttal and revised paper, we focused on **clarifying novelty** and strengthening evidence **through new content and experiments**.
- Differentiation from recent tile‑based DSLs (Reviewers CYSa, CKBd): We clarified how TileLang differs from ThunderKittens, Mojo, Gluon, Tilus, and Helion in both abstraction and design philosophy. We also added **new experiments comparing TileLang with these DSLs**, demonstrating clear advantages in performance and code simplicity.
- Contributions over TVM (Reviewers BJV9, CKBd, 7JBZ): We added a **new architecture overview** of the pipeline (`TileLang DSL → FTG → TensorIR → CUDA/ROCm`) clarifying its tile-level DSL and FTG-based framework atop TVM, and **new experiments comparing TileLang with TVM Ansor** on H100, demonstrating much shorter tuning time and higher FLOPs.
- Automation and speedup sources (Reviewers CYSa, CKBd): We added **pseudocode for tile recommendation and inference**  in the appendix, clarified the boundary between developer control and compiler automation, and adding a **new ablation experiment** that decompose TileLang over the Triton baseline on FlashMLA to explain sources of speedup.
- Scheduling and warp control (Reviewers CYSa, CKBd): We clarified that the pipelined abstraction supports fine‑grained scheduling and warp control, while low‑level primitives remain available for fully manual warp‑specialized kernels.

After these updates, reviewers are broadly satisfied.
- Reviewer BJV9 continues to recommend acceptance.
- Reviewer CKBd states that all concerns are addressed and **raised the score from 8 to 10** before it was reverted to the pre-discussion value.
- Reviewer 7JBZ confirms that the main issues and all questions have been resolved and looks forward to TileLang’s industrial adoption.
- Reviewer CYSa has not yet posted a follow-up, but we provided point-by-point responses to all of his concerns.

TileLang is **open-source** and has already been used to **develop new, efficient AI kernels in production (e.g., DSA in DeepSeek-V3.2-Exp)**. Our practical experience demonstrates its significant value and necessity. We would be grateful if the Area Chair could kindly consider the clarifications along with the practical impact of TileLang during decision-making.

---

### Meta-Review · Area_Chair_91YG · 2026-01-12

**Summary:**

TileLang is a domain-specific programming language for the implementation of fused kernels for deep learning operations on GPUs that focuses on the "tile" (or: a rectangular slice of a multidimensional array) as a primitive in the language and intermediate representation. In doing so it makes the expression of kernels more concise and their execution more efficient by giving more explicit control of memory, data movement and layout, and parallelism along with tools to recommend and infer tiling.

Four expert reviewers with publications and applied experience on deep learning computation, compilers, and GPU implementation strongly recommend acceptance (Bjv9: 8, CKBd: 8, 7JBZ: 8) or marginally recommend rejection (CYSa: 4). The votes for acceptance agree on the importance of the problem, the thorough evaluation across real kernels and both NVIDIA and AMD hardware, and the achieved balance of simple expression and efficient execution. The vote for rejection acknowledges the quality and soundness of the system, but raises concerns about the research contribution in terms of related work and how the algorithms backing TileLang actually work.

The rebuttal is thorough: it provides point-by-point replies to each review, revises the submission, and provides a general summary. Further discussion and revision is provided based on reviewer comments. All points are partially or fully resolved in detail, which includes the addition of new results as requested by review across hardware, DSLs, and different types of kernels and computation (GEMM vs. MHA, irregularity, ...).

The meta-reviewer sides with acceptance, as the majority recommendation of the reviewers, and as the likely consensus were reviews and recommendations to be updated. On inspection of the submission, rebuttal, and revision, the concerns indicating rejection have been addressed and the arguments for acceptance have been confirmed. As further strengths the meta-reviewer highlights that TileLang is open-source software and has been used to develop kernels used in released models, which support the reproducibility and relevance of the system and results in this submission.

**Reviewer Concerns:**

- How TileLang differs from and contributes to work already done by existing languages and toolkits (Triton, Thunderkittens, Mojo, TVM) [CYSa]: The rebuttal clarifies implementation details and adds material to the appendix to describe the differences. In particular, code size reduction is mostly due to tile-level indexing vs. element-level indexing and automatic context and thread handling vs. launching and thread indexing. TileLang is built on top of the backend provided by TVM. The further discussion and additional results show the relationship and performance of TileLang vs. the even more recent DSLs of Tilus, Gluon, and Helion. Resolved.
- How the implementation and performance differ across NVIDIA and AMD GPUs [CYSa]: TileLang is able to share the implementation at its level of abstraction. In the rebuttal and the added Appendix K the shared high-level TileLang code and the resulting platform-specific kernel code are explained. On The AMD MI300X, TileLang achieves 94.8% of the state-of-the-art CK kernel for flash attention, which addresses the specific question of the reviewer. Resolved.
- How TileLang performs on more hardware and in comparison with more recent work [CKBd, CYSa]: While the H100 GPU is popular, the newer Blackwell and older Ampere architectures are also widely used. Gluon and Tilus are concurrent and related alternatives to TileLang that deserve comparison.  Additional results on more hardware (RTX 4090) and with other DSLs (Tilus, Gluon, Helion) show performance improvement. Resolved.
- More fully citing and credit existing work and carefully scoping claims [CKBd, CYSa]: TVM is a foundation for TileLang, more recent DSLs are not cited, and speedup claims may be too cursory in their discussion of the mechanisms for the speedup and how large they are on average. The rebuttal and revision report speedup as a range, to be more conservative as suggested. Resolved.
- How TileLang can schedule advanced cases and what the Pipeline abstraction supports [CYSa, CKBd]: This has been clarified by the rebuttal and expert-level control is still possible through low-level primitives.

**Reviewer Scores:**

- CYSa: the score for marginal rejection (4) would likely increase to at least marginal acceptance (6), because the research concerns raised were addressed by the thorough, point-by-point rebuttal and the content in the revision. The main argument for rejection was lack of clarity in the performance differences and contributions relative to existing recent DSLs, and this information is provided.
- CKBd: the score would go from accept (8) to award (10) based on the review, thorough rebuttal, revision content, and the extensive discussion.
- BJV9: the score of accept (8) would be maintained: the review was already positive and the rebuttal responds to the questions. Reviewer comment confirms the positive evaluation.
- 7JBZ: the score of accept (8) would be maintained or increased: the review was already positive, the rebuttal responds to the questions and includes revisions that incorporate the suggestions, and the reviewer has commented that the weaknesses are addressed (save for one, that is then addressed by the following revision).

---

### Decision · Program_Chairs · 2026-01-26

Accept (Oral)